# One-dimensionally oriented self-assembly of ordered mesoporous nanofibers featuring tailorable mesophases via kinetic control

Liang Peng [1,2,5], Huarong Peng[1,3,5], Steven Wang[2], Xingjin Li [1], Jiaying Mo[2], Xiong Wang[2], Yun Tang[1], Renchao Che [1], Zuankai Wang [2,4,6] ✉, Wei Li [1,6] ✉ & Dongyuan Zhao [1,6] ✉

One-dimensional (1D) nanomaterials have sparked widespread research interest owing to their fascinating physicochemical properties, however, the direct self-assembly of 1D porous nanomaterials and control over their porosity still presents a grand challenge. Herein, we report a monomicelle oriented self-assembly approach to fabricate 1D mesoporous nanostructures with uniform diameter, high aspect ratio and ordered mesostructure. This strategy features the introduction of hexamethylenetetramine as a curing agent, which can subtly control the monomicelle self-assembly kinetics, thus enabling formation of high-quality 1D ordered mesostructures. Meanwhile, the micellar structure can be precisely manipulated by changing the reactant stoichiometric ratio, resulting in tailorable mesophases from 3D cubic (*Im-3m*) to 2D hexagonal (*p6mm*) symmetries. More interestingly, the resultant mesoporous nanofibers can be assembled into 3D hierarchical cryogels on a large scale. The 1D nanoscale of the mesoporous nanofibers, in combination with small diameter (~65 nm), high aspect ratio (~154), large surface area (~452 $m^2 g^{-1}$), and 3D open mesopores (~6 nm), endows them with excellent performances for sodium ion storage and water purification. Our methodology opens up an exciting way to develop next-generation ordered mesoporous materials for various applications.

Low-dimensional nanomaterials have attracted widespread attention owing to their fascinating mechanical, optical, and electrical merits resulted from the reduction of dimensions[1-7]. Over the past years, there is increasing interest in the topological design of nanomaterials with delicate controls on their morphology, structure, and composition[8-14]. As an illustrative example, one-dimensional (1D) nanomaterials offer a number of fascinating advantages because of

their high-aspect-ratio structural feature, combined with high surface area, flexibility, quantum confinement effects, and rich surface functionalities, which enables the formation of a 3D interconnected open texture with low material-consuming but high stretchability[15-22]. These intriguing physical and chemical characteristics endow them huge potential in many high-tech applications, including adsorption, catalysis, sensor, biomedicine, energy conversion and storage, etc[23-30].

[1]Department of Chemistry, Laboratory of Advanced Materials, Shanghai Key Laboratory of Molecular Catalysis and Innovative Materials, iChEM and State Key Laboratory of Molecular Engineering of Polymers, Fudan University, Shanghai 200433, P. R. China. [2]Department of Mechanical Engineering and Research Center for Nature-Inspired Engineering, City University of Hong Kong, Hong Kong 999077, P. R. China. [3]Department of Chemistry, The University of Hong Kong, Hong Kong 999077, P. R. China. [4]Department of Mechanical Engineering, The Hong Kong Polytechnic University, Hong Kong 999077, P. R. China. [5]These authors contributed equally: Liang Peng, Huarong Peng. [6]These authors jointly supervised this work: Zuankai Wang, Wei Li, Dongyuan Zhao. ✉e-mail: zk.wang@polyu.edu.hk; weilichem@fudan.edu.cn; dyzhao@fudan.edu.cn

Additionally, introducing porous structure into 1D nanomaterials can not only provide abundant active sites and interfaces for reaction but also alleviate mechanical stress and accommodate volumetric changes, which are highly desirable for many mass diffusion-limited applications[31,32].

Up to now, tremendous research efforts have been dedicated to developing methodology for synthesizing 1D porous nanomaterials, such as nanocasting[33], self-templating[34,35], hydrothermal carbonization[36,37], self-assembly[38,39], electrospinning[40–42], and biomass conversion[43]. However, most of samples prepared using these approaches limited to small pore size or irregular porosity. As a result, the capacity of the guest materials that can be loaded in the pores and the efficiency of mass/ions transport are quite restricted by their pore size and structure. In this regard, bringing ordered mesoporous structures into 1D nanofibers may achieve optimized, enhanced, or even new-yet-unexpected properties. Several recent studies have indicated that the 1D ordered mesostructures can be achieved by assembling the block copolymers in confined spaces, such as anodic aluminum oxide (AAO) films, or via a nano-casting route to duplicate the porous structure from original 1D templates[44,45]. Such syntheses showed the ability in morphology control but were also known for their obvious shortcomings,

including time-consuming, low yield, and environmental risk. In addition, the obtained materials usually suffered from uncontrollable mesophase. The development of a simple but powerful self-assembly approach templated by amphiphilic block copolymers to simultaneously achieve ordered mesostructure and 1D morphology is highly interesting and desirable. The presence of tunable mesophases within nanofibers, such as 3D open mesoporosity, can further offer broad accessibility for mass transport and significantly improve the loading capacity for guest species[46–48]. Despite significant advances in synthetic methodology, the direct fabrication of ordered mesoporous nanofiber with tailorable mesophases and porosities still remains challenging.

In this work, 1D ordered mesoporous carbonaceous nanofibers (OMCFs) were fabricated via a kinetically driven monomicelle oriented self-assembly strategy by using F127 triblock copolymer (PEO$_{106}$-PPO$_{70}$-PEO$_{106}$) as the structure-directing agent, hexamethylenetetramine (HMT) as the curing agent, and phenolic resol ($M_w$ = 500−2000) as the carbon source under hydrothermal condition, and followed by carbonization in N$_2$ atmosphere (Fig. 1a). Firstly, the low-molecular weight resol oligomers were prepared according to previous report[49], which could form three-connected covalently frameworks by thermopolymerization (Supplementary Fig. 1a). Next, uniform F127/Resol

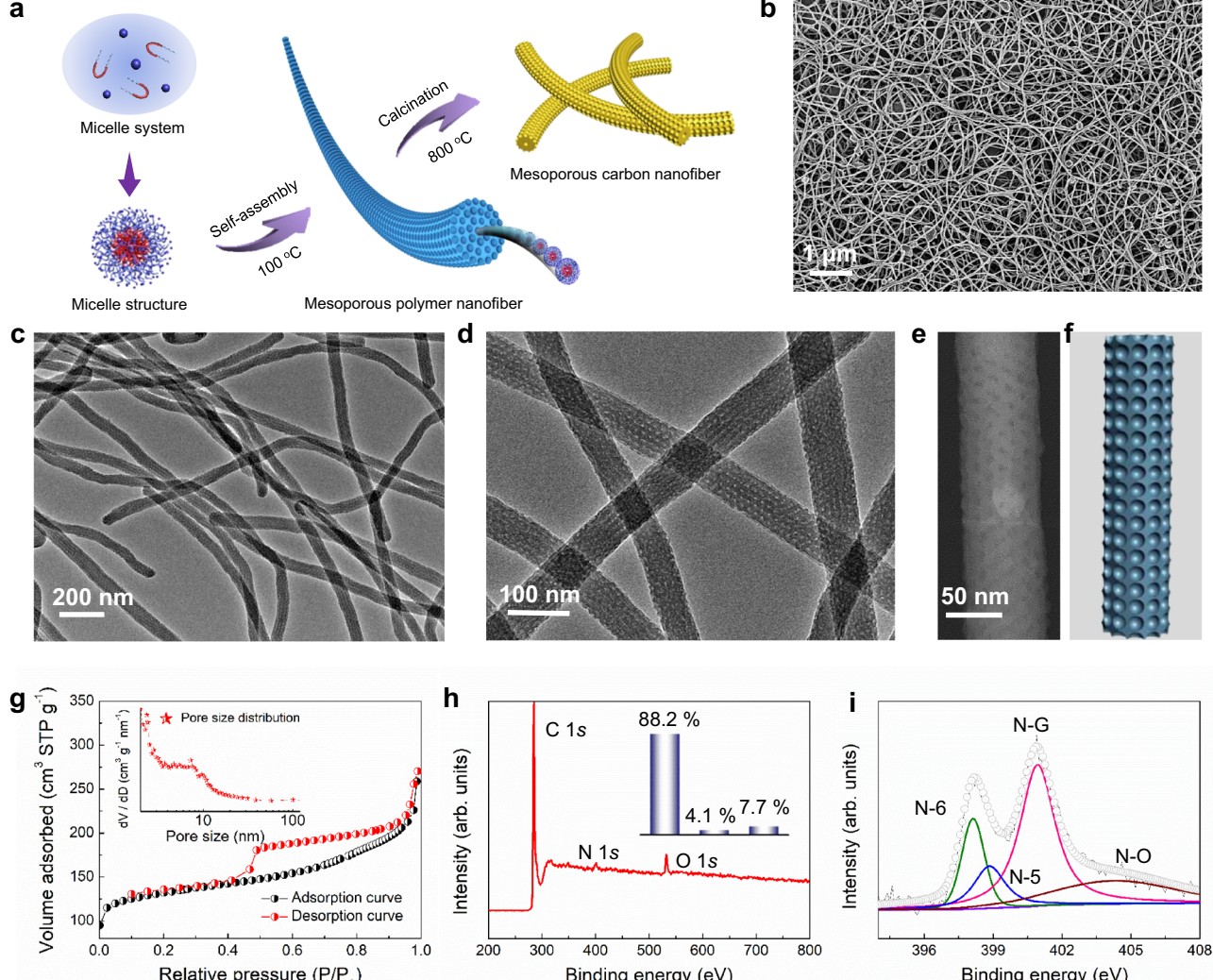

**Fig. 1 | Microstructure characterization of the ordered mesoporous nanofibers.** **a** Schematic illustration of the synthesis process. **b** SEM image. **c**, **d** TEM images with different magnifications. **e** STEM image and (**f**) structural model of the ordered mesoporous nanofibers. **g** N$_2$ sorption isotherms, (**h**) XPS survey and (**i**) high-resolution N 1$s$ spectrum of the cubic OMCFs. Insets of **g** and **h** are the pore size distribution curve and the element weight percentages, respectively.

composite monomicelles system could be formed by stirring of block copolymer F127 molecules and resol oligomers at a rate of 300 rpm (Supplementary Fig. 1b). Then, the composite monomicelles could self-assemble and aggregate with the resol oligomers into ordered mesoporous polymeric nanofibers (OMPFs) under a low-temperature condition (100 °C; Supplementary Fig. 1c). Finally, the freeze-dried OMPFs could be easily transformed into OMCFs via high-temperature carbonization in $N_2$ atmosphere (Supplementary Fig. 1d).

## Results and discussion
### Morphological and structural characterization
The scanning electron microscopy (SEM) image of the as-made OMPFs shows remarkable uniform 1D morphology with an average length of ~10 μm (Supplementary Fig. 2a). Transmission electron microscopy (TEM) image further indicates the existence of 3D open mesoporous structure throughout the OMCFs with a diameter of ~80 nm (Supplementary Fig. 2b). After carbonization in $N_2$ atmosphere, the polymeric nanofibers can be easily converted into mesoporous carbon nanofibers (Fig. 1b). Impressively, the 1D nanofiber morphology is well-preserved without obvious deformation and collapse during the carbonization process (Fig. 1c). The magnified TEM image further confirms that the formed mesoporous structure is well-ordered with a reduced diameter of ~65 nm (Fig. 1d). The aspect ratio of OMCFs is calculated to be ~154. The scanning TEM (STEM) image reveals that the pore size and wall thickness are about ~6 and 7 nm, respectively (Fig. 1e, f). According to the thermogravimetric analysis (TGA) result, the carbon yield is calculated to be ~53% (Supplementary Fig. 3). Two broad peaks are observed in the X-ray diffraction (XRD) pattern at around ~23.8° and 43.9°, which belongs to diffraction modes of hard carbon (Supplementary Fig. 4). The high-resolution TEM image displays that the mesopore walls mainly consist of amorphous carbon structure with randomly oriented lattice fringes (Supplementary Fig. 5). Such amorphous skeleton can be confirmed by Raman spectroscopy (Supplementary Fig. 6), where high $D$ to $G$ band intensity ratio is presented. $N_2$ sorption isotherms of the OMCFs is of type-IV curves with $H_4$ hysteresis loop (Fig. 1g), indicating the copresence of micropores and mesopores in OMCFs. The micropores are mainly originated from the removal of PEO segments from the pore walls and evolution of gases from the organic polymers during carbonization[47,50]. The BET surface area and pore volume are calculated to be ~452 $m^2 g^{-1}$ and 0.43 $cm^3 g^{-1}$, respectively. The pore size distribution is centered at ~6 nm based on Barrett-Joyner-Halenda (BJH) method (inset of Fig. 1g), which is agreement with the TEM results. The X-ray photoelectron spectroscopy (XPS) confirmed only C, N, and O elements in the OMCFs frameworks with the weight percentages of 88.2, 4.1 and 7.7%, respectively (Fig. 1h). The N 1s spectrum reveals that the presence of four forms of nitrogen in the OMCFs, including pyridinic N (398.4 eV), pyrrolic N (399.7 eV), graphitic N (400.8 eV), and oxidized N (402.6 eV; Fig. 1i).

### Formation process and mesophase control
To gain insights into the formation process of the interesting OMCFs, a series of time-dependent control experiments were performed. Initially, through mixing the Pluronic F127 molecules and resol oligomers in the water, a stable and transparent F127/Resol composite monomicelle system can be formed, which shows a typical Tyndall effect under red laser illumination (Fig. 2a). The cryo-TEM image reveals that the F127/Resol monomicelles are uniform with a diameter of about

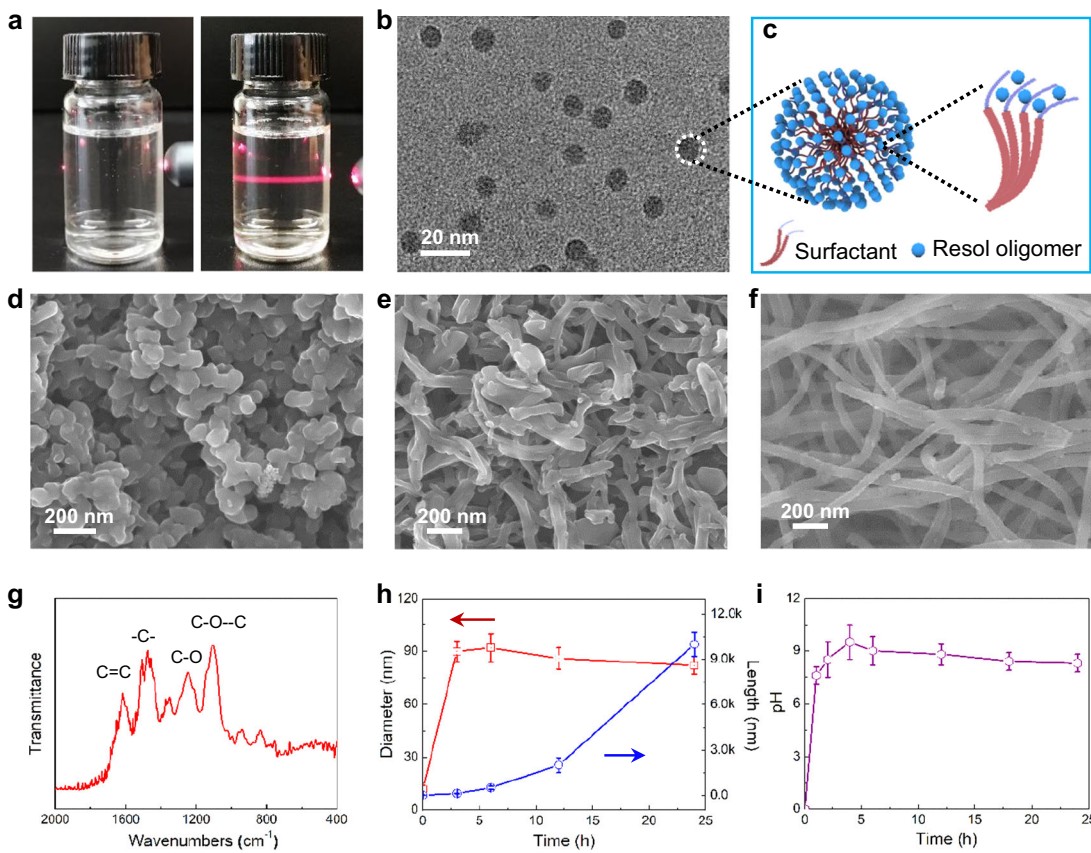

**Fig. 2 | The formation process of the ordered mesoporous nanofibers. a** Optical photographs of the pure water (left) and F127/Resol monomicelle systems (right) under red laser illumination. **b** Cryo-TEM image and **c** corresponding structural model of the monomicelles. SEM images of the nanofibers prepared by the kinetically driven monomicelle oriented self-assembly approach at different reaction times: (**d**) 3 h, (**e**) 6 h, and (**f**) 24 h. **g** FT-IR spectra of the mesoporous polymeric nanofibers. **h** Diameter and length of products, and (**i**) pH value of solution evolutions during the whole formation process.

~12 nm (Fig. 2b). In the spherical monomicelles, the hydrophobic PPO segments of Pluronic F127 surfactants spontaneously aggregate together due to the van der Waals interaction and are surrounded by hydrophilic PEO parts to form the core-shell structure (Fig. 2c). The resol oligomers can interact with the hydrophilic PEO segments of Pluronic F127 surfactants through hydrogen bonding. After introduction of HMT into the system, the F127/Resol monomicelles would aggregate and self-assembly into mesoporous nanoellipsoid firstly (Fig. 2d). As the reaction progressed, more monomicelles were continuously assembled at the ends of the nanoellipsoid through an oriented self-assembly mode to generate the mesoporous nanorods (Fig. 2e), and finally achieving uniform mesoporous nanofibers (Fig. 2f). The FT-IR spectrum of the mesoporous polymeric nanofibers show two absorption peaks at 1612 and 1463 $cm^{-1}$, which assigns to the C = C aromatic ring and -CH$_2$- deformation vibration (Fig. 2g). The peak at 1249 $cm^{-1}$ is corresponded to the phenolic -OH in-pane deformation. A clear peak at 1056 $cm^{-1}$ is indexed to the C-O-C deformation vibration, indicating the presentation of F127 molecules in the nanofibers. The length of the sample increases gradually form ~ 100 nm to 10 μm as the self-assembly progresses, while the diameter almost remains constant (Fig. 2h). Meanwhile, the pH value of the solution gradually increased to 9.5 in the initial 3 h due to the thermally induced decomposition of HMT molecules, and then remained unchanged (Fig. 2i), demonstrating the buffering role of HMT molecules.

In order to understand the 1D self-assembly mechanism, the effect of catalyst was firstly studied. When using ammonia (NH$_3$·H$_2$O) as a catalyst, uniform carbonaceous nanofibers can be formed but without ordered mesoporous structure (Supplementary Fig. 7a), which is confirmed by small angle X-ray scattering (SAXS) analysis (Supplementary Fig. 7b). Replacing the catalyst with NaOH solution, no precipitates can be obtained at different kinds of concentrations (Supplementary Fig. 7c). Further changing the catalyst to HMT, uniform nanofibers are achieved with well-defined mesostructure (Supplementary Fig. 7d). The variability in the products confirms that the used catalyst has a strong influence of the monomicelle self-assembly kinetics. Compared with ammonia and NaOH, HMT molecules can in-situ release of ammonia into the solution and thus dynamically mediate the monomicelle self-assembly kinetics to achieve 1D oriented self-assembly. Notably, the HMT concentration also affects the morphology and structure of the resultant products (Supplementary Fig. 8). By increasing the HMT concentration from 0.2 to 0.8 g L$^{-1}$, the formed products can be continually changed from nonporous nanoparticle to mesoporous nano-beans, and then to ordered mesoporous nanofibers (Supplementary Fig. 8a–c). Once a high concentration of HMT is used (1.6 g L$^{-1}$), the ordered mesostructure is disappeared and irregular solid particle is appeared (Supplementary Fig. 8d). The structural evolution of products is probably because the HMT concentration can remarkably influence the self-assembly behavior between the resol oligomers and F127/Resol monomicelles.

Meanwhile, the effects of reactant concentration and pH value on morphology and pore structure of products were investigated. For convenience to describe, the reactant concentration used in the Methods section is set to 100 %. When the reactant concentration is 50%, no precipitation is formed probably because the micelle concentration is lower than its critical micelle concentration (Supplementary Fig. 9a). By continuously increasing the reactant concentration from 75% to 100%, the structure can be tuned from nonporous nanoparticles to mesoporous nanofibers (Supplementary Fig. 9b, c). Further increasing the reactant concentration to 125 %, irregular solid nanoparticles are appeared again (Supplementary Fig. 9d). This is because the reactant concentration can significantly affect the nucleation rate of the resol oligomers and monomicelle self-assembly kinetics[51]. Alternatively, the importance of pH for the formation of 1D mesoporous structure has been further confirmed. At a low pH value of ~ 5, aggregated solid nanoparticles with irregular shape

are obtained (Supplementary Fig. 10a). By increasing the pH value from ~ 8 to 9.5, the structure of products can be varied from interconnected porous nanoribbons to mesoporous nanofibers (Supplementary Fig. 10b, c). However, no precipitation is formed as the pH value increases up to ~ 12 (Supplementary Fig. 10d). These results indicate that the weakly alkaline environment is favorable for achieving kinetic balance between the resol oligomers polymerization and F127/Resol monomicelles self-assembly to generate 1D morphology and mesoporous structure[52].

Furthermore, the effects of temperature and surfactant were also investigated. When the reaction temperature is increased from 80 to 100 °C, the structure of the products can be altered from mesoporous nano-beans to mesoporous nanofibers (Supplementary Fig. 11a, b). On further increasing the temperature to 120 °C, hollow nanorods are appeared (Supplementary Fig. 11c). When the temperature is increased up to 140 °C, interconnected particles are formed without porous structure (Supplementary Fig. 11d). It is indicated that the reaction temperature applied can significantly influence the decomposition rate of HMT molecules, thus changing the self-assembly behavior of F127/Resol monomicelle. Meanwhile, too high reaction temperature can lead the monomicelle unstable and even broken[53,54]. In addition, the F127/Resol mass ratio also greatly affect the shape and pore structure of the products (Fig. 3). At a mass ratio of 0.30, the SAXS pattern shows three well-resolved scattering peaks at 0.48, 0.67, and 0.83 that assigned to the 110, 200, and 211 scattering reflections of 3D cubic mesophase (Im-3m) (Fig. 3a), which is further confirmed by the SEM and TEM images (Fig. 3b, c). By carefully increasing the F127/Resol mass ratio from 0.35 to 0.40, the spherical pores are gradually expanded and evolved into cylindrical structure within the nanofibers (Fig. 3d–i). Interestingly, the mesoporous nanofibers become curved and crosslinked with the increase of F127/Resol mass ratio (Fig. 3e, h). When the mass ratio reaches 0.45, the SAXS result shows three shifted peaks at 0.45, 0.78, and 0.90, which are ascribed to the 100, 110, and 200 reflections of 2D hexagonal mesophase (p6mm) (Fig. 3j–l). The TEM image further shows that the hexagonal OMCFs possess ordered mesoporous channels parallel to the long axis, and their diameter is ~ 85 nm (Supplementary Fig. 12a). The channel diameter is about ~ 8 nm. The EDX mapping further displays that the OMCFs are composed of C, N, and O elements (Supplementary Fig. 12b), suggesting the heteroatoms-doped functionalization. The BET surface areas of the obtained mesoporous nanofibers are calculated to be ~ 452, 421, 347, and 264 m$^2$ g$^{-1}$, respectively (Supplementary Fig. 13). The corresponding pore size distributions are centered at ~ 6.0, 6.8, 7.4, and 8.0 nm, respectively. All data are summered in Supplementary Table 1. The scheme illustrates the mesoporous structure evolution of the OMCFs prepared using different F127/Resol mass ratios (Fig. 3m). However, when the absence of Pluronic F127 or using a low F127/Resol mass ratio (0.10) in the system, only nanospheres are formed (Supplementary Fig. 14a, b). Once the F127/Resol mass ratio is increased up to 0.60, twisted nanorods are appeared with irregularly cylindrical structure (Supplementary Fig. 14d).

## Kinetically driven one-dimensional self-assembly mechanism

Based on the above observations, we propose a kinetically driven monomicelle-oriented self-assembly mechanism to illustrate the synthesis of the above-mentioned OMCFs with precisely tailorable ordered mesophases (Fig. 4). In general, the morphological control of mesoporous nanomaterials achieved through block copolymers self-assembly is greatly dependent on the subtle molecular interactions among the reacting species. Specially, the hydrogen-bonding is the most important factor for the preferential organization of the organic precursors with the hydrophilic blocks of amphipathic templates. In our system, the low-molecular weight resol and amphiphilic Pluronic F127 triblock copolymer are selected as the organic precursor and structure-directing agent, respectively, which can form hydrogen-

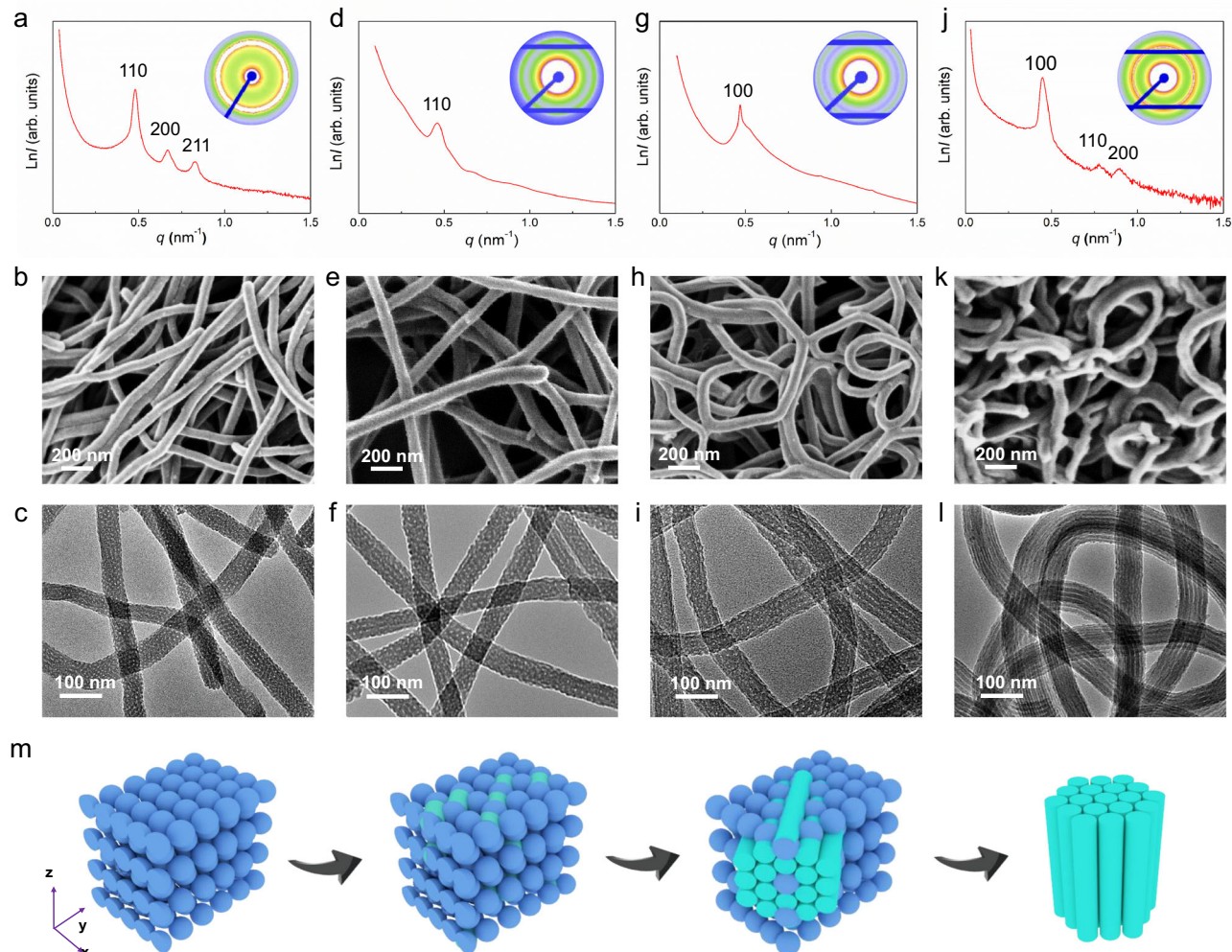

**Fig. 3 | Different mesophases of the ordered mesoporous nanofibers.** SAXS patterns, SEM images, and TEM images of the OMCFs prepared by the kinetically driven monomicelle oriented self-assembly approach using different F127/Resol mass ratios: (**a–c**) 0.30, (**d–f**) 0.35, (**g–i**) 0.40, and (**j–l**) 0.45. The synthetic processes were performed similarly to the "Methods" section in the Supplemental Information, except that the F127/Resol mass ratio was altered. **m** The corresponding structural models.

bonding networks between hydroxyl groups of phenols and PEO moieties (Fig. 4a). As agitation is adopted, the shearing force can induce the micellization of Pluronic F127 surfactants into core-shelled spherical structure with PPO domains as the core and surrounded by resol associated PEO shells, forming F127/Resol composite monomicelle as the *meso*-building block (Supplementary Fig. 15). The key feature of our synthesis is the use of HMT molecules as a mediator and curing agent because these can in-situ decompose into formaldehyde and ammonia molecules at an elevated temperature. On the one hand, the as-derived ammonia molecules can serve as a pH buffer to intelligently control the self-assembly kinetics of monomicelles on demand. On the other hand, the formaldehyde molecules can act as a mediator to manipulate the molecular interaction between the resol oligomers through hydrogen bonding, that is, to regulate their polymerization kinetics. Consequently, reasonable control of the reaction temperature with adoption of an appropriate HMT concentration might allow to finely balance the kinetics between monomicelle self-assembly and resol polymerization, so as to realize morphological and structural controls (Fig. 4b). In detail, a low reaction temperature (mode I, 80 °C) endows both of the F127/Resol monomicelles and resol oligomers with a low reaction kinetics due to the slow rate of HMT hydrolysis. Thus, the F127/Resol monomicelles could self-assemble and aggregate slowly with the resol oligomers to form the ordered mesoporous nano-

beans (Supplementary Fig. 15a). Increasing the reaction temperature (mode II, 100 °C) is conducive to the hydrolysis process of HMT molecules, which promotes the in-situ release of ammonia and formaldehyde molecules in the solution, and ultimately resulting in faster polymerization rate between resol oligomers. More importantly, that will switch the polymerization mode of resol oligomers from 3D network to linear style, leading to achieve a kinetic balance between the monomicelles self-assembly and resol precursors polymerization and finally resulting in the 1D mesoporous nanostructure (Fig. 4c and Supplementary Fig. 15b). In this case, the spherical F127/Resol monomicelles are firstly self-assembled into mesoporous short nanorods through a body-centered cubic mode (Step 1a). As the assembly progresses, more monomicelles are continuously oriented grown on the ends of the nanorods, namely, epitaxially self-assembled along the long-axial direction, thus generating the longer mesoporous nanofibers. The mesoporous nanofibers are finally solidified by thermal polymerization of phenol-formaldehyde polymers (Step 1b). After the carbonization in $N_2$ atmosphere, the polymeric framework can be easily converted into carbon one, resulting in 1D OMCFs with 3D cubic symmetry (*Im-3m*) (Step 1c). The corresponding morphological evolution is confirmed by the time-dependent experiments (Fig. 2). Only nonporous nanofiber or solid particle can be obtained when using pure ammonia or formaldehyde as the curing agents (Supplementary

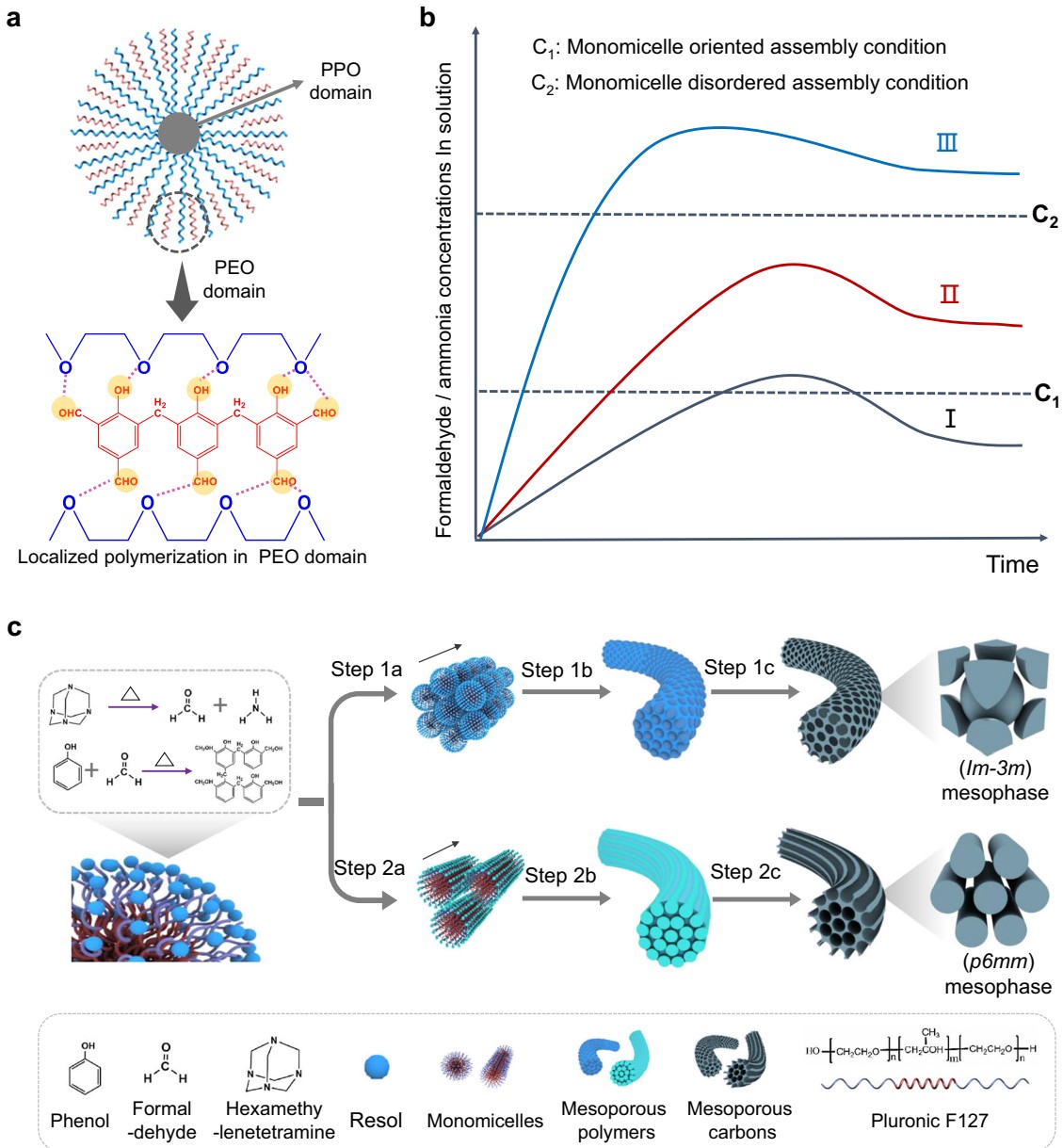

**Fig. 4 | Kinetically driven mechanism for the monomicelle one-dimensional oriented self-assembly. a** Structural characteristic of the F127/Resol monomicelle with PPO domains as the core and surrounded by resol associated PEO shells. **b** Schematic illustrations of the formaldehyde and ammonia concentrations in the solution varied with the reaction time under different reaction temperatures (mode I: 80 °C; mode II: 100 °C; mode III: 140 °C). **c** Schematic illustration of the fabrication of cubic OMCFs (Route1) and hexagonal OMCFs (Route 2) using the kinetically driven monomicelle oriented self-assembly approach.

Figs. 7a and 16). These results indicate that only the in-situ release of ammonia and formaldehyde molecules decomposed form HMT can form ordered 1D mesoporous structure. More importantly, the F127/ Resol mass ratio can greatly affect the mesophase of the formed OMCFs (Fig. 3). This is because increasing the amphipathic Pluronic F127 mass ratio would expand the monomicelle size and subsequentially decrease the curvature of the micellar surface to minimize the interfacial energy, which allows the monomicelles to form cylindrical nanorods rather than nanospheres[49,55]. As a result, the adjacent micelles would undergo a touching, merging, and fusing process along the self-assembly direction to form a cylindrical-like geometry, bringing about the formation of mesoporous nanofibers with parallel nanochannels (Step 2a, b). After the freeze-drying and carbonization processes, uniform OMCFs were generated with a 2D hexagonal symmetry (*p6mm*) (Step 2c). However, a higher F127/Resol mass ratio could cause a change in micellar morphology and consequently induce

mesoporous nanofibers to be bend and increase in dimension (Supplementary Fig. 14d). In addition, further increasing the reaction temperature (mode III, 140 °C) could break down the overall kinetic equilibrium between the monomicelles self-assembly and resol oligomers polymerization (Supplementary Fig. 15c), resulting in nonporous solid particles. Therefore, well-defined 1D mesostructures with precisely tunable mesophases can be facilely synthesized through a monomicelles self-assembly kinetic control pathway, which could be promising for diverse applications.

### Fabrication of the 3D hierarchical cryogel
Interestingly, the OMCFs can be assembled into three-dimensional (3D) hierarchical porous cryogel on a large-scale through a unidirectional ice-templating strategy[18,56] (Fig. 5a). The optical photograph shows that the morphology and structure of polymeric cryogel can be well-maintained after calcination at 800 °C (Fig. 5b). The SEM images

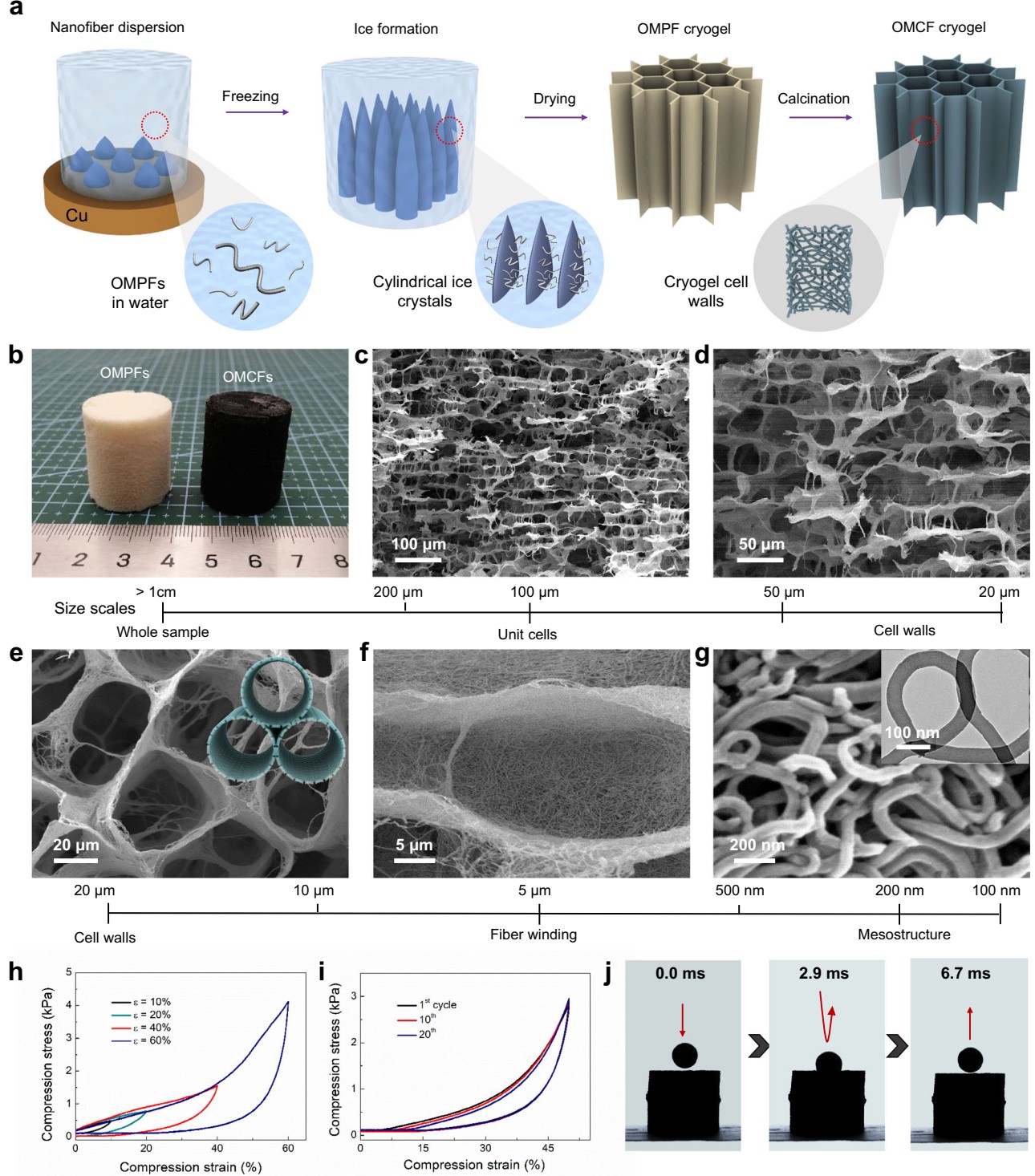

**Fig. 5 | Structural design and fabrication of the 3D nanofibers cryogel.**
**a** Schematic illustration, (**b**) optical photograph and (**c–g**) SEM images of the
OMCFs cryogel. **h** Compressive σ-ε curves of the OMCFs cryogel with increasing ε
amplitude. **i** Cyclic fatigue test with ε = 50 %. **j** Optical photographs of plastic ball
rebound experimnt. Insets of **e** and **g** are their corresponding structural model and
TEM image, respectively.

show that the carbon cryogel is consisted of hierarchical cellar struc-
ture with large-scale parallel channels (Fig. 5c), which is similar to the
structure of the nature wood. The channels are linked with each other
by stochastic bridges, and the gap between adjacent walls is about
~40 μm (Fig. 5d). The top view of the cryogel shows a honeycomb-like
scaffold paralleling to the z direction (Fig. 5e). The thickness of the cell
walls are only few hundred nanometers (Fig. 5f). Magnified SEM image
further reveals that the OMCFs are intertwined and bonded with each

other to form the cell walls (Fig. 5g), endowing the 3D cryogel with
good mechanical stability. The dynamic compressive stress-strain
(σ–ε) tests show that the as-built OMCFs cryogel can completely
recover their original shape up to the stain of ε = 60% (Fig. 5h), sug-
gesting the durable hyperelasticity. The maximum σ could reach up to
4.2 kPa under an applied ε of 60%. Furthermore, the multicycle com-
pression measurements exhibit that the OMCFs cryogel can retain η = ~
96% of the initial maximum σ after 20 cycles (Fig. 5i). When being

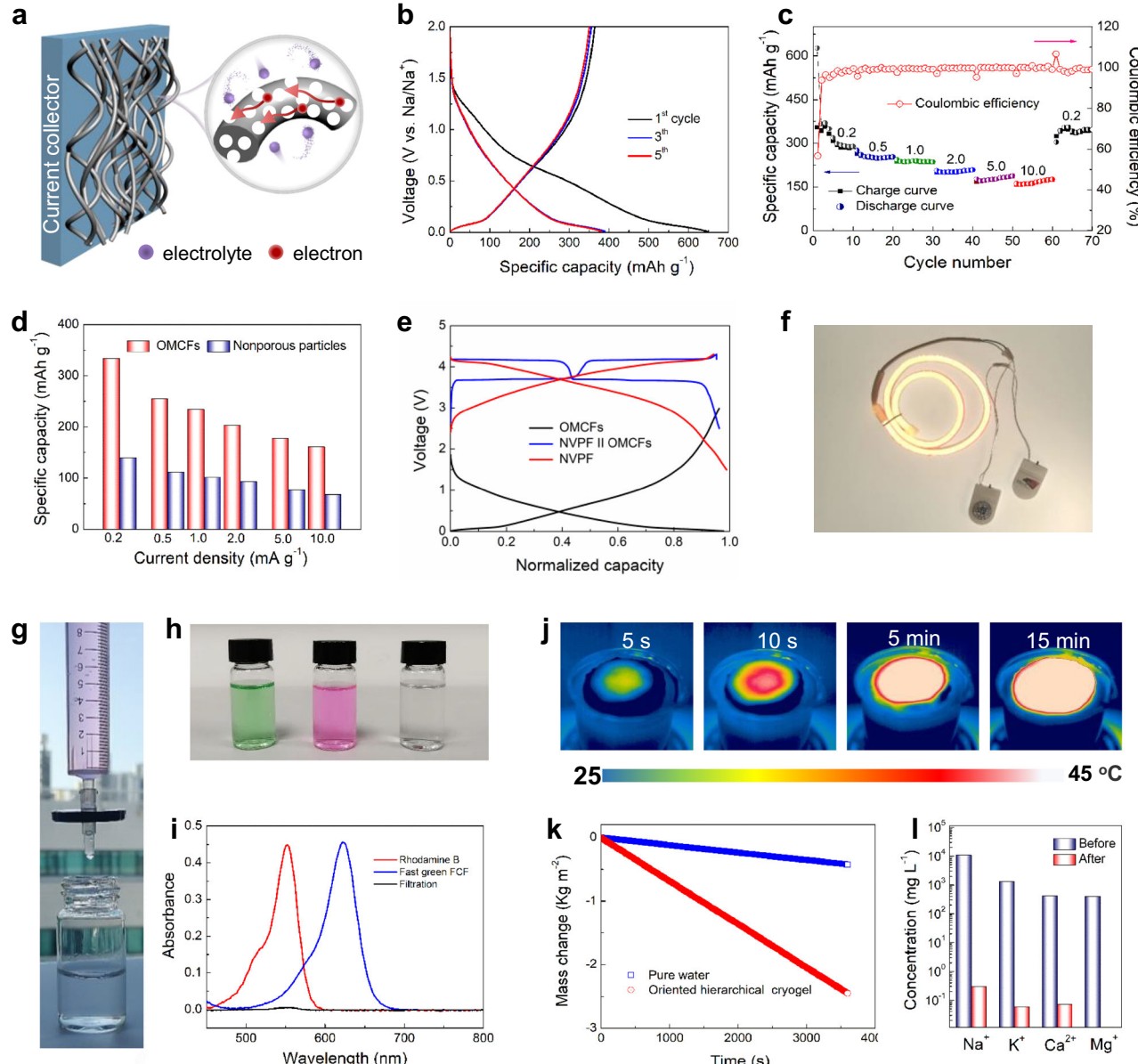

**Fig. 6 | Diverse applications of the ordered mesoporous nanofibers and their derived cryogel. a** Schematic illustration of the electrode composed by the OMCFs for sodium-ion storage. **b** Galvanostatic charge/discharge curves at 0.1 A g⁻¹ and **c** rate capability and coulombic efficiency at various current rates from 0.2 to 10.0 A g⁻¹ of the OMCFs electrode. **d** Comparison of rate capabilities between OMCFs and nonporous particles electrodes. **e** Charge/discharge curves of the OMCFs anode, Na₃V₂(PO₄)₃F₃ cathode, and OMCFs ‖ Na₃V₂(PO₄)₃F₃ sodium-ion full battery, respectively. **f** Optical photograph of a LED strip lamp powered by two full batteries in series. **g** Optical photograph of the nanofilter device based on OMCFs cryogel. **h** Optical photograph and **i** UV-vis spectra of the solutions before and after filtering Rhodamine B and fast green FCF. **j** IR thermal images of the OMCFs cryogel evaporator under one-sun illumination. **k** Mass change curves of the water on OMCFs cryogel evaporator with pure water as control. **l** The concentrations of four primary ions in seawater before and after purifying.

compared with randomly linked nanofibers cryogels, the oriented OMCFs cryogels exhibit greater maximum stress and narrower hysteresis loop (Supplementary Fig. 17), indicating good mechanical robustness. That can be confirmed by the plastic ball rebound experiment (Fig. 5j).

## Diverse applications

Owing to their 1D morphology, well-defined mesostructure, and favorable N-doped properties, the resultant OMCFs cryogel was recognized as an advanced material for practical applications. The electrochemical performance of the cubic OMCFs cryogel was firstly investigated in sodium ion batteries (SIBs; Fig. 6a). The cyclic voltammetry (CV) curves exhibit one irreversible cathodic peak at ~0.7 V may attribute to the formation of a solid-electrolyte interface (SEI)

layer during the initial cycle (Supplementary Fig. 18). A pair of sharp redox peaks at 0 - 0.2 V result from insertion/extraction of Na ion in the carbon frameworks. The galvanostatic charge/discharge profiles show that the initial charge and discharge capacities are ~652 and 388 mAh g⁻¹, respectively, at the current density of 0.1 A g⁻¹ (Fig. 6b). In the subsequent cycles, the OMCF electrode exhibits excellent cyclability with a stable capacity of 346 mAh g⁻¹ after 100 cycles (Supplementary Fig. 19), which is much better than that of the non-porous carbon particle electrode (Supplementary Fig. 20). Such cycling performance is better than that of many other carbon-based electrodes reported previously (Supplementary Table 2). The pore structure stability of OMCFs can be confirmed by TEM analysis after a long-term cycle (Supplementary Fig. 21). Furthermore, the OMCFs electrode delivers reversible capacities of 295, 247, 234, 203, 174, and

163 mAh g$^{-1}$ when the current density increased from 0.2, 0.5, 1.0, 2.0, 5.0–10.0 A g$^{-1}$, respectively (Fig. 6c). Interestingly, the capacity can recover to 343 mAh g$^{-1}$ as the current density returns to 0.2 A g$^{-1}$. In terms of rate capability, the OMCFs electrode is better to nonporous carbon electrode at different current densities (Fig. 5d and Supplementary Fig. 22). In addition, SIBs in full-cell configuration (FIBs) were further assembled by employing the OMCFs cryogel as an anode material and Na$_3$V$_2$(PO$_4$)$_2$F$_3$ (NVPF) as a cathode material, respectively. The galvanostatic charge/discharge profiles of the FIBs shows two working plateaus at ~3.5 (discharge status) and 3.8 V (charge status), respectively, and delivers a specific capacity of 105 mAh g$^{-1}$ at 0.1 A g$^{-1}$ based on the NVPF cathode material (Fig. 6e). In addition, two coin-type FIBs could power a commercial LED lamp (Fig. 6f), indicating its great potential for practical applications.

The advantages of 1D morphology and 3D open mesostructure of the OMCFs became more prominent for water purification such as dye separation and seawater desalination. A nanofilter device was firstly assembled by using OMCFs cryogel as active layer in a dismountable filter holder (Fig. 6g and Supplementary Fig. 23). To evaluate the performance of the filtration device for dye separation, two different organic dyes (Rhodamine B and Fast green FCF) were employed as examples. Optical photographs clearly show that the two organic dyes have distinct red and green colors in aqueous solutions (Fig. 6h). After filtration, the UV-vis spectrum reveals that the characteristic peaks of organic dyes at 552 and 623 nm almost completely disappeared (Fig. 6i), indicating great filtering performance of the nanofilter device. That can be confirmed by time-dependent adsorption tests (Supplementary Fig. 24). The OMCF cryogels possess an impressive adsorption capacity of 384 mg g$^{-1}$ for Rhodamine B, which is higher than that of the commercial activated carbon. Alternatively, the OMCF cryogel also can act as a promising material to assemble into vapor device for efficient seawater desalination (Supplementary Fig. 25). IR thermal images indicate that the surface temperature of the device rises from 23 to ~42 °C in 5 min under simulated one-sun irradiation (100 mW cm$^{-2}$) (Fig. 6j). In contrast, the surface temperature of pure water keeps nearly unchanged after 15 min of illumination (~25 °C) (Supplementary Fig. 26). The water mass changes with and without OMCFs cryogel were collected once the devices achieved a steady temperature (Fig. 6k). It is clear that the vapor device with hierarchical OMCFs cryogels presents a high evaporation rate of 2.4 kg m$^{-2}$ h$^{-1}$, which is ~ 6 and 1.5 times that for pure water and randomly linked nanofibers, respectively (Supplementary Fig. 27). The evaporation performance is comparable with that of many materials reported previously (Supplementary Table 3). Furthermore, four types metal ions (namely, Na$^+$, Mg$^{2+}$, K$^+$, and Ca$^{2+}$) in the original seawater source show 4 orders of concentration decrement after desalination (Fig. 6l), which are well below the WHO standard for safe drinking water.

The exceptional performance of OMCFs for practical application can be attributed to their fascinating nanostructure and chemical properties, including nanoscale dimension, 1D morphology, high aspect ratio, 3D ordered open mesostructure, amorphous framework, and rich O, N-dopants. Firstly, the nanoscale dimension and 3D ordered open mesostructure of OMCFs offer much shorter migration distances for mass transfer in comparison with bulk materials, allowing organic molecules and metal ions to easily penetrate into the whole nanomaterials. Secondly, the 1D morphology and high aspect ratio of OMCFs can construct hierarchical porous networks integrating nanoscale and the microscale morphologies effects, which not only provides a 3D continuous pathway for mass transfer and electron transport but also offers multi-level pore structures and large stacking space to buffer the mechanical stress/strain and volume change during reaction processes. Thirdly, the big pore size, thin pore walls, and amorphous frameworks can offer large confined spaces, accessible surfaces/interfaces, and abundant active sites for guest molecule storage and reaction. Fourthly, the homogeneous rich N, O-dopants in OMCFs can further enhance the electronic conductivity and tailor interfacial wettability, thus additionally advance their performance.

In summary, we have demonstrated a simple but efficient mono-micelle oriented self-assembly approach to fabricate the OMCFs with well controlled 1D morphology and mesoporous structures. In this synthesis, we have shown the possibility of modulating the mono-micelle self-assembly kinetics through a HMT hydrolysis induced process, which provides an opportunity to simultaneously realize 1D morphology and ordered mesostructure controls. By regulating the reaction parameters such as the reactant stoichiometric ratio, it is convenient to systemically tailor the ordered mesophase of the prepared OMCFs from 3D cubic (*Im-3m*) to 2D hexagonal (*p6mm*) symmetries. The resultant OMCFs show small diameter (~65 nm), high aspect ratio (~154), large surface area (~452 m$^2$ g$^{-1}$), 3D large open mesopores (~6 nm), and favorable nitrogen dopants (~4.1 wt%). Impressively, the OMCFs can be assembled into hierarchical porous cryogel on a large scale. As a result, the OMCFs and their derived 3D cryogels deliver excellent rate capability (163 mAh g$^{-1}$ at 10.0 A g$^{-1}$) for SIBs and great water purification performance (2.4 kg m$^{-2}$ h$^{-1}$) for seawater desalination, respectively. We believe that this work presents flexible and versatile tools to prepare interesting 1D ordered mesoporous nanomaterials for various advanced applications and, more impressively, gains insights into the monomicelle self-assembly chemistry and science.

## Methods

### Synthesis of ordered mesoporous carbon nanofibers

The OMCFs were synthesized by a kinetically driven monomicelle oriented self-assembly strategy (see Supplementary Information for the more detailed synthesis procedure). In a typical process, 1.0 mL of resol precursor solution, 0.24 g of Pluronic F127, and 0.08 g of HMT were added into 80 mL of deionized water under stirring at 300 rpm to obtain the F127/Resol monomicelle system. The stirring bar was 2 cm in length. Then, the above mixture was placed into a 100 mL Teflon-lined stainless-steel autoclave for hydrothermal treatment at 100 °C for 24 h. After the autoclave was cooled, the as-made polymeric composites were collected by centrifugation and washing with water and ethanol. Finally, the freeze-dried polymeric composites were further calcined at 800 °C for 2 h under N$_2$ atmosphere to result in the OMCFs with well-defined mesostructure.

### Synthesis of 3D OMCFs cryogel

The hierarchical OMCFs cryogel was prepared through an ice-templating method. Typically, 1.0 g of ordered mesoporous polymeric nanofibers were dispersed in 300 mL of deionized water by stirring at 10000 rpm for 30 min, yielding uniform nanofiber dispersion. Next, the obtained dispersion was transferred into a designed Teflon mold, which was subsequently frozen in a liquid nitrogen tank and then dried for 48 h to generate the hierarchical polymeric nanofibers (OMPFs) cryogel. Finally, the OMPFs cryogel was carbonized at 800 °C for 2 h to produce the OMCFs cryogel.

## Data availability

Data supporting the findings of this study are included within the article and Supplementary Information files. Data are available from the authors upon request.

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

## Acknowledgements

This work was supported by the National Key R&D Program of China (Nos. 2022YFA1503501 (W.L.) and 2018YFA0209401 (D.Z.)), National Nature Science Foundation of China (Nos. 22105041 (D.Z.), 21733003 (D. Z.), U21A20329 (W.L.), 51975502 (D.Z.), and 21975050 (W.L.)), Program of Shanghai Academic Research Leader (No. 21XD1420800 (W.L.)), Shanghai Pilot Program for Basic Research-Fudan University 21TQ1400100 (No. 21TQ008 (W.L.)), and Science and Technology Commission of Shanghai Municipality (No. 22JC1410200 (D.Z.)). Research Grants Council of Hong Kong (Nos. SRFS2223–1S01 (Z.W.), C1006-20W (Z.W.), and 11219219 (Z.W.)), and Innovation and Technology Commission of Hong Kong (Nos. GHP/021/19SZ (Z.W.) and GHP/092/20GD (Z.W.)).

## Author contributions

L.P., W.L., and D.Z. conceived the project and designed the experiments. L.P., Z.W., W.L., and D.Z. co-wrote the manuscript. L.P. carried out the synthesis and characterization of the materials. H.P. involved in the application data collection. X.L., S.W., J.M., X.W., Y.T., and R.C. assisted L.P. for the data collection and analysis. All authors contributed to the discussion and manuscript preparation.

## Competing interests

The authors declare no competing interests.
