## [Peer Review File · Nature Communications]

One-dimensionally oriented self-assembly of ordered mesoporous nanofibers featuring tailorable mesophases via kinetic controlREVIEWER COMMENTS

Reviewer #1 (Remarks to the Author):

The manuscript “One-dimensionally oriented self-assembly of ordered mesoporous nanofibers featuring tailorable mesophases via kinetic control” provides a new and versatile strategy for producing long, mesoporous polymer (and carbon) nanofibers with ordered mesostructures. The technical quality of the manuscript is high, however some control experiments are missing for supporting some of the conclusions.

Major points:

- 1) The role of pH was not tested for controlling and altering the self-assembly, however a tremendous effect is expected on both the kinetics and the final (quasi-equilibrium) state of the self-assembly. This could also shed some light on the role of H-bonding and other interactions that drive the micelle and other superstructure formation. The fundamental roles of such solution phase effects are discussed in length in the “Kinetically driven one-dimensional self-assembly mechanism” chapter, but pH specific experimental evidence is not provided.
- 2) The detailed method of the ice templating / controlled freezing should be given, because this procedure is fundamental in determining the final morphology of the produced cryogels. This in-turn will have a significant effect in their application related performance. Without experimental controls in ice formation, the conclusions of the “Fabrication of the 3D hierarchical aerogel” chapter are not supported.
- 3) As a continuation of the previous point; the role of the morphologies of the OMCFs cryogels are completely overlooked in the “Diverse applications” chapter. Both the nanoscale and the microscale morphologies have well-defined and complex effects in electrochemistry, mass-transport and sorption; that is why they are needed to be controlled.
- 4) As there are many materials prepared by varying the catalyst / temperature / HMT concentration / F127-Resol ratio, each figure caption both in the main text and in the Supporting Information should explicitly identify the exact preparation conditions of the materials shown in each figure panel. This is needed to properly guide and inform the reader.

Minor points:

- 5) The ice-template macroporous cryogels cannot be termed as “aerogels”. This is misleading for the

readers.

6) The last paragraph of the Introduction is a summary of the results of the present work (and repeats the Abstract). This is not needed for an introduction.

7) There should be a statement in the main text that the experimental details enabling the reproduction of the materials are given in the Supporting Information.

Reviewer #2 (Remarks to the Author):

In this work, Zhao et. al. report an interesting monomicelle oriented self-assembly approach to fabricate mesoporous nanofibers with uniform diameter, high aspect ratio and ordered mesostructure. Detailed control experiments indicate that the monomicelle self-assembly behavior is followed a kinetic control pathway, which provides a new opportunity to simultaneously realize ordered mesostructure and 1D morphology controls. This concept has never been reported and it will make a new direction in the research of 1D materials. Furthermore, the micellar structure in this system can be delicately adjusted by altering the reactant stoichiometric ratio, resulting in diverse novel ordered mesostructures. The quality of the data of the manuscript is high level and the materials were well characterized with the sophisticated facilities to prove their concept. Therefore, I suggest accepting this work for publication in Nat. Commun. after following issues are taken into account.

1. Besides the catalyst and reaction temperature, other factors, such as the reactant concentration and molar ratio may also affect the micellar chemistry, thus affecting the morphology and mesostructure of the resultant samples.

2. If it is possible to control the diameter and length of the synthesized nanofibers?

3. Organic molecules (e.g., hexamethylenetetramine, HMT) not only play an important role in the evolution of morphology but also significantly affect the interfacial interaction between soft templates and precursors. More comments are required.

4. How about the materials' stability? This point is very critical if we consider practical applications. It is better to add more comments on this, especially the pore structure.

5. Although the performance shown in this manuscript is very good, it is still recommended to have a summary and comparison with other samples published in previous papers or commercial products.

Reviewer #3 (Remarks to the Author):

The manuscript “One-dimensionally oriented self-assembly of ordered mesoporous nanofibers featuring tailorable mesophases via kinetic control” demonstrates an interesting self-assembly based 1D carbon material design strategy screening systematically parameters and deriving observation-based design rules. The manuscript includes mechanistic conclusions (Figure 4) as well as a material application-oriented section. The main innovative part of the manuscript, to my opinion, can be seen in the material fabrication section. Unfortunately, this section remains partly superficial and statements are only partly supported by data and characterization data is in parts incomplete. For example, the apparently present micropores (Figure 1g) are completely ignored in the discussion? In general, the manuscript writing remains in part imprecise and contains errors. For example in figure captions images are not correctly assigned, so that not all statements are supported or linked to data and that it is in part hard to clearly review figures and statements. The second part of the manuscript aiming to demonstrate application potential remains weak as neither clear influence of controllable structure on the material performance are shown nor is an improved performance demonstrated as compared to classic carbon materials derived.

To my opinion major revisions are suggested and a transfer to another material fabrication specified journal may be considered, in case the mechanistic part is not more systematically supported by data, and / or the structure property relationship and related advantage of 1D structure on material performance is not clearly transferred into the stated application performance or related to a performance increase. Regarding the first point I suggest to systematically and not only exemplarily combine gas adsorption, TEM and SAXS data for structure analysis and relating observations to the process parameters and solution composition.

I try to specify my suggestions below: In Figure 1 the authors show nitrogen adsorption data and their pore size distribution analysis. The pore size distribution shows mesopores (IUPAC: 2-50 nm) as well as micropores (IUPAC < 2nm) as far as this can be deduced from the x-axes and adsorption isotherm. This

should be included into the discussion as well as the exact analysis of this data (models used etc.). The authors only refer to mesopores within the entire manuscript. The micropore formation should be included into the mechanistic discussion (Figure 4), too. In addition, it has to be considered that the presence of micropores would probably significantly increase the specific surface area. Furthermore, the measurement seems not to be finished as the desorption branch is not measured back to low relative pressure. Why is this the case? In this regard as well Figure S11 has to be explained as adsorption and desorption branch are not coming into contact again in the low partial pressure regime. The authors relate the isotherm type to a type IV (I assume a type Iva according to IUPAC) isotherm. This is not fully consistent as the adsorption branch shows a very slow increase and the desorption shows pore blocking and micropores. In addition, another increase at very high relative pressure occurs and should be explained. Thus, the adsorption data interpretation should be reconsidered and gas adsorption results should be added for all relevant porous structures not only for the one example in Figure 1. It should be especially added to Figure 3, to complement TEM and SAXS data. To my opinion, this would improve and solidify the interpretation of porous structure formation together with SAXS data in Figure 3 as well as the mechanistic insights discussed in Figure 4. Furthermore, according to the IUPAC report from 2015 (Pure Appl. Chem. 2015; 87(9-10): 1051–1069), it is known that nitrogen adsorption bears a relatively large error with respect to pore size determination, especially with respect to varying surface chemistry. IUPAC thus recommends argon adsorption. Especially carbon materials are known to adsorb species from the environment and thus change their surface chemistry. Why did the authors use nitrogen and not argon adsorption measurements and why only for two selected examples?

The authors use the term “F127/HMT/Resol monomicelles”. This term should be defined. Subsequently, the authors state, that these “monomicelles”, which seem to be detrimental for the desired material formation, are formed at a stirring rate of 300 rpm. To make this value reproducible more information about the process such as about the flask / reactor / volumes etc. is needed. A pure stirring rate is not useful. All information should be provided and ideally the process understanding beyond a pure value for a stirring rate under very specific conditions should be provided. The experimental section should be reconsidered with respect to providing all needed information. Figure S1 for example suggests that the hydrothermal reaction was conducted in simple glass vials with blue cap. Is this really the case?

Figure S5 does not clearly show ordered mesoporosity? Could the authors explain this image clearly

and relate to the scale bar? In several figure captions in the SI the authors state cubic order of mesopores. Which data supports cubic mesopore arrangement? I do neither see SAXS data showing clear mesopore order in these Figures nor a relation to SAXS data shown elsewhere. In Figure 3 SAXS data is shown but not in all Figures in which a specific order is proposed. On the other hand, gas adsorption data is missing in Figure 3. SAXS and gas sorption data should be added e.g. to Figure S8 for example, too. The statements regarding structural properties should be carefully re-checked and either corrected or supported or linked to the corresponding data. For example, Figure S8 does not even indicate the presence mesopore in the first and last image. In addition, the assignment of figure letters in Figure S8 (a, b, c, d) does not seem to be correct. This makes it in general very difficult to consistently review this manuscript and connect processing parameters and resulting structural properties. Characterization data should be completed, Figures should be correctly assigned and statements should be clearly connected to data. The figure caption in Figure 2 does not fully correspond to the Figure content. It is difficult to understand especially images h and i. For example, what is displayed at the right y-axis in h and how was this value determined? The right y-axis description is not visible in the pdf as it is overlaid by the Figure i. To give one further example: In the discussion referring to Figure 2 the authors state at page 8: "well-defined mesoporous structure (Fig. 2f)." From the depicted SEM images no mesopore structure can be deduced. The entire paragraph should be reconsidered and corrected. Statements should be supported and linked to the corresponding data.

Regarding the proposed mechanism (Figure 4): On which data or references is the resol location within the micelles based on? Could the authors make this clear in the manuscript? The observed structural data should be reconsidered with this mechanism e.g. including the role of micropore formation? At page 13 the authors state: "On increasing the temperature (mode II, 100 °C) enables the increase of hydrogen-bonding interaction between the resol oligomers and monomicelles". This seems counter intuitive to me. Could the authors specify this statement? Following up on the 1D structure formation, the authors report on aerogel formation by an "ice-templating strategy" and specifically describe the resulting structure. Could the authors comment on reproducibility of this aerogel structure and resulting material characteristics such as the capacity or adsorption properties shown in Figure 6? I assume the resulting aerogel structure sensitively depends on the process conditions. Furthermore, the described material properties such as dye adsorption are not very surprising but rather reflecting well-known carbon material properties. Could

the authors comment on specific advantages over carbon-based materials prepared via alternative and well-known strategies? Is the adsorption capacity significantly increased for example? I cannot deduce such a performance increase which could be related to the 1D structure from the given data. The authors indicate such improved performance e.g. by the statement: "...exhibits excellent cyclability with a stable capacity of 346 mAh g⁻¹ after 100 cycles (Supplementary Fig. S16), which is much better than that of the nonporous carbon particle electrode (Supplementary Fig. S17), indicating the morphological and structural advantages of the OMCs.". But no reference to the state of the art and most recent literature is given and a clear structure property relationship is not deduced. Consequently, this statement remains vague and imprecise. In a second example on water filtration the authors state: "peaks of organic dyes at 552 and 623 nm almost completely disappeared (Fig. 6i), indicating superior filtering performance of the nanofilter device.". Without giving adsorption capacity and comparison to benchmark materials this statement does not allow to deduce and superior performance. Furthermore, which functional groups are responsible for high adsorption capacity in case the material exists of almost pure carbon? In general, none of the applications, which are all known for classic carbon materials are clearly related to the 1D material design. In general this application-related section remains weak with respect to innovative performance or structure driven insights but rather shows expected behavior of carbon-based materials.

Title: “One-dimensionally oriented self-assembly of ordered mesoporous nanofibers featuring tailorable mesophases via kinetic control”

Point-to-Point Response to Reviewers

Reviewer #1:

The manuscript “One-dimensionally oriented self-assembly of ordered mesoporous nanofibers featuring tailorable mesophases via kinetic control” provides a new and versatile strategy for producing long, mesoporous polymer (and carbon) nanofibers with ordered mesostructures. The technical quality of the manuscript is high, however some control experiments are missing for supporting some of the conclusions.

Response: We appreciate the reviewer for the positive comments.

Comment 1. *The role of pH was not tested for controlling and altering the self-assembly, however a tremendous effect is expected on both the kinetics and the final (quasi-equilibrium) state of the self-assembly. This could also shed some light on the role of H-bonding and other interactions that drive the micelle and other superstructure formation. The fundamental roles of such solution phase effects are discussed in length in the “Kinetically driven one-dimensional self-assembly mechanism” chapter, but pH specific experimental evidence is not provided.*

Response: We appreciate the reviewer for this useful suggestion. We have accepted it and carried out the pH control experiments in the revised manuscript (Supplementary Fig. 10). The pH of the reaction solution was adjusted by diluted HCl (0.6 M) and NaOH (1 M) solutions. At a low pH value of ~ 5, aggregated solid nanoparticles with irregular shape are obtained (Supplementary Fig. 10a). By increasing the pH value from ~ 8 to 9.5, the structure of products can be varied from interconnected porous nanoribbons to mesoporous nanofibers (Supplementary Fig. 10b, c). However, no precipitation is formed as the pH value increases up to ~ 12 (Supplementary Fig. 10d). These results indicate that the pH indeed plays a critical role in the formation of 1D morphology and mesoporous structure.

On Page 9, Line 16, in the revised manuscript, we have added: “**Alternatively, the importance of pH for the formation of 1D mesoporous structure has been further confirmed. At a low pH value of ~ 5, aggregated solid nanoparticles with irregular shape are obtained (Supplementary Fig. 10a). By increasing the pH value from ~ 8 to 9.5, the structure of products can be varied from interconnected porous nanoribbons to mesoporous nanofibers (Supplementary Fig. 10b, c). However, no precipitation is formed as the pH value increases up to ~ 12**

(Supplementary Fig. 10d). These results indicate that the pH indeed plays a critical role in the formation of 1D morphology and mesoporous structure.”

Accordingly, in the revised Supplementary Information, we have added a new figure and caption: “**Supplementary Fig. 10.** The TEM images of the carbon materials prepared by the kinetically driven monomicelle oriented self-assembly approach using different pH values: (a) ~5, (b) 8, (c) 9.5 and (d) 12. The pH of the reaction solution was adjusted by diluted hydrochloric acid (0.6 M) or potassium hydroxide (1.0 M) solutions. The synthetic processes were conducted similarly to the above Supplementary Method section except that the pH value was varied. The polymeric samples were converted into carbon ones through a carbonization process at 800 °C for 2 h in N₂ atmosphere.”

Comment 2. *The detailed method of the ice templating / controlled freezing should be given, because this procedure is fundamental in determining the final morphology of the produced cryogels. This in-turn will have a significant effect in their application related performance. Without experimental controls in ice formation, the conclusions of the “Fabrication of the 3D hierarchical aerogel” chapter are not supported.*

Response: We appreciate the reviewer for the useful suggestion. We have accepted it and provided more details about the ice templating method in the revised manuscript. Meanwhile, we also have prepared randomly linked OMCFs cryogels and compared them with oriented hierarchical OMCFs cryogels. The synthetic process was performed similarly with oriented hierarchical OMCFs cryogels except that the aqueous nanofiber dispersion was poured in a plastic pipe and followed by freezing in a refrigerator (-20 °C). The result shows that the oriented OMCFs have greater maximum stress and narrower hysteresis loop than the randomly linked nanofibers cryogels (Supplementary Fig. 17), indicating good mechanical robustness.

On Page 4, Line 16, in the Methods section of the revised Supplementary Information, we have added more details: “The hierarchical OMCFs cryogels were prepared through an ice-templating method. In a typical synthesis, 1.0 g of ordered mesoporous polymeric nanofibers (OMPFs) was dispersed in 300 mL of deionized water by stirring at 1000 rpm for 30 min, yielding uniform nanofiber dispersion with a concentration of 3.4 mg mL⁻¹. Then, 15 ml above mixing dispersion was transferred to a designed cylindrical Polytetrafluoroethylene (PTFE) mold (2.5 cm in diameter, 3.5 cm in depth), where the bottom was covered by a copper gasket (0.2 cm in thick). Next, the whole mold was immersed in liquid nitrogen to generate a temperature gradient from bottom to top. After the dispersion was completely frozen, the device was placed in a freeze-dryer more than 24 h under 1.0 Pa pressure and -80 °C to

generate the OMPFs cryogels. Finally, the OMPF cryogels were pre-heated at 350 °C for 3 h and further kept at 800 °C for 2 h with a heating rate of 1 °C min⁻¹ in N₂ atmosphere to form the OMCFs cryogels.

The randomly linked OMCFs cryogels were prepared by the similar synthetic procedure except that the aqueous polymeric nanofiber dispersion was poured in a plastic pipe and followed by freezing in a refrigerator (-20 °C) for 24 h.”

On Page 17, Line 3, in the revised manuscript, we have added: “When being compared with randomly linked nanofibers cryogels, the oriented OMCFs cryogels exhibit greater maximum stress and narrower hysteresis loop (Supplementary Fig. 17), indicating good mechanical robustness.”

Accordingly, in the revised Supplementary Information, we have added a new figure and caption: “**Supplementary Fig. 17.** Stress-strain curves of (a) randomly linked nanofibers cryogels and (b) oriented OMCFs cryogels under a high strain compression, respectively.”

Comment 3. *As a continuation of the previous point; the role of the morphologies of the OMCFs cryogels are completely overlooked in the “Diverse applications” chapter. Both the nanoscale and the microscale morphologies have well-defined and complex effects in electrochemistry, mass-transport and sorption; that is why they are needed to be controlled.*

Response: We appreciate the reviewer for the useful suggestion. We have accepted it and conducted the structural control experiments. Firstly, we have investigated the water evaporation performances of randomly linked and oriented hierarchical OMCFs cryogels. The result shows that the hierarchical OMCFs cryogel device exhibits a high evaporation rate of 2.4 kg m⁻² h⁻¹, which is ~1.5 times that for randomly linked OMCFs one (Supplementary Fig. 27). Secondly, in terms of sodium ion storage, the OMCFs electrode is superior to nonporous carbon electrode at different current densities (Supplementary Fig. 22). Thirdly, the hierarchical OMCFs cryogels possess an impressive adsorption capacity of 384 mg g⁻¹ for Rhodamine B, which is higher than that of commercial activated carbon. All of these data indicate that the resultant oriented hierarchical structure has a significant effect in their application related performance. In addition, we have systematically summarized the advantages of the hierarchical OMCFs cryogels in the revised manuscript.

We have accepted the suggestion, on Page 19, Line 19, in the revised manuscript, we have added: “That can be confirmed by time-dependent adsorption tests (Supplementary Fig. 24). The OMCF cryogels possess an impressive adsorption capacity of 384 mg g⁻¹ for Rhodamine B, which is higher than that of commercial activated carbon.”

On Page 20, Line 5, in the revised manuscript, we have added: “It is clear that the vapor device with hierarchical OMCFs cryogels present a high evaporation rate of $2.4 \text{ kg m}^{-2} \text{ h}^{-1}$, which is ~ 6 and 1.5 times that for pure water and randomly linked nanofibers, respectively (Supplementary Fig. 27).”

On Page 17, Line 18, in the revised manuscript, we have added: “Such cycling performance is better than that of many other carbon-based electrodes reported previously (Supplementary Tab. 2).”

On Page 19, Line 3, in the revised manuscript, we have described: “In terms of rate capability, the OMCFs electrode is superior to nonporous carbon electrode at different current densities (Figure 5d and Supplementary Fig. 22).”

On Page 20, Line 12, in the revised manuscript, we have added: “The exceptional performance of OMCFs for practical application can be explained with their unique 1D morphology, ordered mesostructure and rich N-doped chemical property. First, the nanoscale diameter and 1D morphology of OMCFs provide much shorter migration distances for mass transportation in comparison with bulk materials, allowing organic molecules and ions to easily penetrate into the whole nanomaterials. Second, the ultra-long OMCFs can form hierarchical webs integrating nanoscale and the microscale morphologies effects, which not only provides a 3D continuous pathway for mass transport but also offers multi-level pore structures and large stacking space to buffer the mechanical stress/strain and volume change during reaction processes. Third, the 3D open ordered mesoporous structure and thin pore walls can offer plentiful confined spaces, surfaces/interfaces and active sites for guest molecule storage and reaction. Fourth, the homogeneous N, O dopants in OMCFs can further alter the electronic conductivity and interfacial wettability, thus additionally contributing to their performance improvement.”

Accordingly, in the revised Supplementary Information, we have added two new figures and captions: “**Supplementary Fig. 24.** Time-dependent adsorption capacities of Rhodamine B onto the ordered mesoporous carbon nanofibers and commercial activated carbon.”

“**Supplementary Fig. 27.** (a) Mass change curves of the water on hierarchical OMCFs cryogel and randomly linked nanofibers cryogel evaporators under one-sun illumination. (b) Schematic illustration showing the water transportation of the above-mentioned two solar-driven evaporators.”

Comment 4. *As there are many materials prepared by varying the catalyst / temperature / HMT concentration / F127-Resol ratio, each figure caption both in the main text and in the Supporting Information should explicitly identify the exact preparation conditions of the materials shown in each figure panel. This is needed to properly guide and inform the reader.*

Response: We appreciate the reviewer for the useful suggestions. We have accepted them and added more preparation details in figure panels both in the Main text and in the Supplementary Information.

Comment 5. *The ice-template macroporous cryogels cannot be termed as “aerogels”. This is misleading for the readers.*

Response: We thank the reviewer for the useful suggestion. We have accepted it and replaced the word “aerogels” with “cryogel” in the revised manuscript.

Comment 6. *The last paragraph of the Introduction is a summary of the results of the present work (and repeats the Abstract). This is not needed for an introduction.*

Response: We appreciate the reviewer for the useful suggestion. We have accepted it and deleted the corresponding paragraph in the revised Introduction.

Comment 7. *There should be a statement in the main text that the experimental details enabling the reproduction of the materials are given in the Supporting Information.*

Response: We appreciate the reviewer for the useful suggestion. We have accepted it and added a statement in the revised Method section.

On Page 21, Line 20, in the revised manuscript, we have added: “see Supplementary Information for the more detailed synthesis procedure”

Reviewer #2:

In this work, Zhao et. al. report an interesting monomicelle oriented self-assembly approach to fabricate mesoporous nanofibers with uniform diameter, high aspect ratio and ordered mesostructure. Detailed control experiments indicate that the monomicelle self-assembly behavior is followed a kinetic control pathway, which provides a new opportunity to simultaneously realize ordered mesostructure and 1D morphology controls. This concept has never been reported and it will make a new direction in the research of 1D materials. Furthermore, the micellar structure in this system can be delicately adjusted by altering the reactant stoichiometric ratio, resulting in diverse novel ordered mesostructures. The quality of the data of the manuscript is high level and the materials were well characterized with the sophisticated facilities to prove their concept. Therefore, I suggest accepting this work for publication in Nat. Common. after following issues are taken into account.

Response: We thank the reviewer for the positive comments.

Comment 1. *Besides the catalyst and reaction temperature, other factors, such as the reactant concentration and molar ratio may also affect the micellar chemistry, thus affecting the morphology and mesostructure of the resultant samples.*

Response: We appreciate the reviewer for the useful suggestion. We have accepted them and carried out the control experiments in term of reactant concentration and mass ratio (Supplementary Fig. 9 and 14). For convenience to describe, the reactant concentration used in the experimental section is set to 100 %. When the reactant concentration is 50 %, no precipitation is formed because the micelle concentration is lower than the critical micelle concentration (Supplementary Fig. 9a). By continuously increasing the reactant concentration from 75 % to 100 %, the structure can be tuned from nonporous nanoparticles to mesoporous nanofibers (Supplementary Fig. 9b, c). Further increasing the reactant concentration to 125 %, irregular solid nanoparticle appeared again (Supplementary Fig. 9d). However, when the absence of Pluronic F127 or using a low F127/Resol mass ratio (0.10) in the system, only nonporous nanospheres are formed (Supplementary Fig. 14a, b). Once the F127/Resol mass ratio is increased up to 0.60, twisted nanorods are appeared with irregularly cylindrical structure (Supplementary Fig. 14d).

On Page 9, Line 8, in the revised manuscript, we have added: “For convenience to describe, the reactant concentration used in the Method section is set to 100 %. When the reactant concentration is 50 %, no precipitation is formed because the micelle concentration is lower than the critical micelle concentration (Supplementary Fig. 9a). By continuously increasing the reactant concentration from 75 % to 100 %, the structure can be tuned from nonporous nanoparticles to mesoporous nanofibers (Supplementary Fig. 9b, c). Further increasing the reactant concentration to 125 %, irregular solid nanoparticles are appeared again (Supplementary Fig. 9d). One reason for the variability in the structure of the products may be that the reactant concentration greatly influences the monomicelle self-assembly kinetics.”

On Page 11, Line 19, in the revised manuscript, we have described: “However, when the absence of Pluronic F127 or using a low F127/Resol mass ratio (0.10) in the system, only nanospheres are formed (Supplementary Fig. 14a, b). Once the F127/Resol mass ratio is increased up to 0.60, twisted nanorods are appeared with irregularly cylindrical structure (Supplementary Fig. 14d).”

Accordingly, in the revised Supplementary Information, we have added a new figure and caption.

“**Supplementary Fig. 9.** The TEM images of the carbon materials prepared by the kinetically driven monomicelle oriented self-assembly approach using different reactant concentrations: (a) 50 %, (b) 75 %, (c) 100 % and (d) 125 %. Here, the reactant concentration used in the Method section is set to 100%. The synthetic processes were conducted similarly to the above Supplementary Method section except that the reactant concentration was varied. The polymeric samples were converted into carbon ones through a carbonization process at 800 °C for 2 h in N₂ atmosphere.”

Comment 2. *If it is possible to control the diameter and length of the synthesized nanofibers?*

Response: We appreciate the reviewer for this useful suggestion. By tuning the reaction time, we can control the length of the synthesized nanofibers from ~ 100 nm to 10 μm, while the diameter almost remains constant during the synthesis process (Fig. 2h).

We have accepted the suggestion of the reviewer. On Page 7, Line 16, in the revised manuscript, we have added the description: “The length of the sample increases gradually from ~ 100 nm to 10 μm as the self-assembly progresses, while the diameter almost remains constant (Fig. 2h).”

Comment 3. *Organic molecules (e.g., hexamethylenetetramine, HMT) not only play an important role in the evolution of morphology but also significantly affect the interfacial interaction between soft templates and precursors. More comments are required.*

Response: We appreciate the reviewer for this useful suggestion. The key feature of our synthesis is the use of HMT molecule as a mediator and curing agent because it is subject to slow hydrolysis into formaldehyde and ammonia at an elevated temperature, enabling precise control of the monomicelle self-assembly kinetics to form 1D ordered mesoporous nanostructure. Specifically, the as-derived ammonia molecule can serve as a pH buffer to intelligently control the self-assembly kinetics of monomicelles on demand. Simultaneously, the formaldehyde molecules can act as a mediator to manipulate the molecular interaction between resol oligomers and PEO domains of monomicelles through hydrogen bonding, that is, to regulate the polymerization kinetics. Consequently, reasonable control of the reaction temperature with adoption of an appropriate HMT concentration might allow to finely balance the reaction kinetics between monomicelle self-assembly and resol polymerization, so as to realize morphological and structural controls

We have accepted the suggestion of the reviewer. On Page 13, Line 14, in the revised manuscript, we have added the description as follows: “The key feature of our synthesis is the use of HMT molecule as a mediator and curing agent because it is subject to slow hydrolysis into formaldehyde and ammonia at an elevated temperature. Specifically, the as-derived ammonia molecule can serve as a pH buffer to intelligently control the self-assembly kinetics of monomicelles on demand. Simultaneously, the formaldehyde molecules can act as a mediator to manipulate the molecular interaction between resol oligomers and PEO domains of monomicelles through hydrogen bonding, that is, to regulate the polymerization kinetics. Consequently, reasonable control of the reaction temperature with adoption of an appropriate HMT concentration might allow to finely balance the kinetics between monomicelle self-assembly and resol polymerization, so as to realize morphological and structural controls (Fig. 4b).”

Comment 4. *How about the materials' stability? This point is very critical if we consider practical applications. It is better to add more comments on this, especially the pore structure.*

Response: We appreciate the reviewer for the useful suggestion. We have accepted it and analyzed the cycled sample using TEM technology (Supplementary Fig. 21). The result shows that the one-dimensional morphology and mesoporous structure can be well retained after the cycling test, which is consistent with its excellent electrochemical durability.

On Page 17, Line 20, in the revised manuscript, we have added: “The pore structure stability of OMCFs can be confirmed by TEM analysis after a long-term cycle (Supplementary Fig. 21).”

Accordingly, in the revised Supplementary Information, we have added a new figure and caption. “**Supplementary Fig. 21.** The TEM image of the ordered mesoporous carbon nanofibers after 100 cycles at a current density of 0.1 A g⁻¹.”

Comment 5. *Although the performance shown in this manuscript is very good, it is still recommended to have a summary and comparison with other samples published in previous papers or commercial products.*

Response: We appreciate the reviewer for the useful suggestion. We have accepted it and summarized some recently reported papers about the carbon-based materials for sodium ion storage and seawater desalination (Supplementary Tab. 2 and Tab. 3). Meanwhile, we have also investigated the adsorption capacities of ordered mesoporous nanofibers and commercial activated carbon for Rhodamine B, respectively (supplementary Fig. 24).

On Page 17, Line 18, in the revised manuscript, we have added: “Such cycling performance is better than that of many other carbon-based electrodes reported previously (Supplementary Tab. 2)”

On Page 19, Line 20, in the revised manuscript, we have added: “The OMCF cryogel possesses an impressive adsorption capacity of 384 mg g⁻¹ for Rhodamine B, which is higher than that of the commercial activated carbon.”

On Page 20, Line 8, in the revised manuscript, we have added: “The evaporation performance is comparable with that of many materials reported previously (Supplementary Tab. 3)”

Accordingly, in the revised Supplementary Information, we have added one figure, two new tables, and corresponding references: “**Supplementary Fig. 24.** Time-dependent adsorption capacities of Rhodamine B onto the ordered mesoporous carbon nanofibers and commercial activated carbon.”

“**Supplementary Table 1.** Comparison of the electrochemical performances of the ordered mesoporous carbon nanofiber electrode in this work with some carbon-based electrodes reported in the literature.”

“**Supplementary Table 2.** Comparison of the water evaporation rate of the ordered mesoporous carbon nanofiber cryogel in this work with some carbon-based evaporators reported in the literature.”

Supplementary References

3. Xia JL, et al. Hard carbon nanosheets with uniform ultramicropores and accessible functional groups showing high realistic capacity and superior rate performance for sodium-ion storage. *Adv. Mater.* **32**, 2000447 (2020).
4. Li Y, Kong M, Hu J, Zhou J. Carbon-microcuboid-supported phosphorus-coordinated single atomic copper with ultrahigh content and its abnormal modification to Na storage behaviors. *Adv. Energy Mater.* **10**, 2000400 (2020).
5. Tang Z, et al. Electrode-Electrolyte Interfacial Chemistry Modulation for Ultra-High Rate Sodium-Ion Batteries. *Angew. Chem. Int. Ed.* **61**, e202200475 (2022).
6. Hu X, et al. Nitrogen-rich hierarchically porous carbon as a high-rate anode material with ultra-stable cyclability and high capacity for capacitive sodium-ion batteries. *Nano Energy* **56**, 828-839 (2019).
7. Fang H, et al. Dual-Function Presodiation with Sodium Diphenyl Ketone towards Ultra-stable Hard Carbon Anodes for Sodium-Ion Batteries. *Angew. Chem. Int. Ed.* **62**, e202214717 (2023).
8. Hou B-H, et al. Self-Supporting, Flexible, Additive-Free, and Scalable Hard Carbon Paper Self-Interwoven by 1D Microbelts: Superb Room/Low-Temperature Sodium Storage and Working Mechanism. *Adv. Mater.* **31**, 1903125 (2019).
9. Li Z, et al. Defective Hard Carbon Anode for Na-Ion Batteries. *Chem. Mater.* **30**, 4536-4542 (2018).
10. Hong Z, et al. Rational Design and General Synthesis of S-Doped Hard Carbon with Tunable Doping Sites toward Excellent Na-Ion Storage Performance. *Adv. Mater.* **30**, 1802035 (2018).
11. Xu Z-L, et al. Tailoring sodium intercalation in graphite for high energy and power sodium ion batteries. *Nat. Commun.* **10**, 2598 (2019).
12. Zhou X, et al. Three-Dimensional Ordered Macroporous Metal-Organic Framework Single Crystal-Derived Nitrogen-Doped Hierarchical Porous Carbon for High-Performance Potassium-Ion Batteries. *Nano Lett.* **19**, 4965-4973 (2019).
13. Huang S, et al. Boosting Surface-Dominated Sodium Storage of Carbon Anode Enabled by Coupling Graphene Nanodomains, Nitrogen-Doping, and Nanoarchitecture Engineering. *Adv. Funct. Mater.* **32**, 2203279

(2022).

14. Zhu L, et al. Self-Contained Monolithic Carbon Sponges for Solar-Driven Interfacial Water Evaporation Distillation and Electricity Generation. *Adv. Energy Mater.* **8**, 1702149 (2018).

15. Xia Y, et al. Spatially isolating salt crystallisation from water evaporation for continuous solar steam generation and salt harvesting. *Energy Environ. Sci.* **12**, 1840-1847 (2019).

16. Wang W, et al. Simultaneous production of fresh water and electricity via multistage solar photovoltaic membrane distillation. *Nat. Commun.* **10**, 3012 (2019).

17. Chen G, et al. Biradical-Featured Stable Organic-Small-Molecule Photothermal Materials for Highly Efficient Solar-Driven Water Evaporation. *Adv. Mater.* **32**, 1908537 (2020).

18. Zhao H-Y, et al. Lotus-inspired evaporator with Janus wettability and bimodal pores for solar steam generation. *Cell Rep. Phys. Sci.* **1**,100074 (2020).

19. Wu L, et al. Highly efficient three-dimensional solar evaporator for high salinity desalination by localized crystallization. *Nat. Commun.* **11**, 521 (2020).

20. Ren L, et al. Designing Carbonized Loofah Sponge Architectures with Plasmonic Cu Nanoparticles Encapsulated in Graphitic Layers for Highly Efficient Solar Vapor Generation. *Nano Lett.* **21**, 1709-1715 (2021).

21. Liu F, et al. Low cost, robust, environmentally friendly geopolymer-mesoporous carbon composites for efficient solar powered steam generation. *Adv. Funct. Mater.* **28**, 1803266 (2018).

22. Zhao L, et al. Shape-programmable interfacial solar evaporator with salt-precipitation monitoring function. *ACS Nano* **15**, 5752-5761 (2021).

23. Xia D, et al. Tuning the Electrical and Solar Thermal Heating Efficiencies of Nanocarbon Aerogels. *Chem. Mater.* **33**, 392-402 (2021).

24. Wang X, et al. An interfacial solar heating assisted liquid sorbent atmospheric water generator. *Angew. Chem. Int. Ed.* **131**, 12182-12186 (2019).

25. Singh SC, et al. Solar-trackable super-wicking black metal panel for photothermal water sanitation. *Nat. Sustain.* **3**, 938-946 (2020).”

Reviewer #3:

The manuscript “One-dimensionally oriented self-assembly of ordered mesoporous nanofibers featuring tailorable mesophases via kinetic control” demonstrates an interesting self-assembly based 1D carbon material design strategy screening systematically parameters and deriving observation-based design rules. The manuscript includes mechanistic conclusions (Figure 4) as well as a material application-oriented section. The main innovative part of the manuscript, to my opinion, can be seen in the material fabrication section. Unfortunately, this section remains partly superficial and statements are only partly supported by data and characterization data is in parts incomplete. For example, the apparently present micropores (Figure 1g) are completely ignored in the discussion? In general, the manuscript writing remains in part imprecise and contains errors. For example in figure captions images are not correctly assigned, so that not all statements are supported or linked to data and that it is in part hard to clearly review figures and statements. The second part of the manuscript aiming to

demonstrate application potential remains weak as neither clear influence of controllable structure on the material performance are shown nor is an improved performance demonstrated as compared to classic carbon materials derived.

To my opinion major revisions are suggested and a transfer to another material fabrication specified journal may be considered, in case the mechanistic part is not more systematically supported by data, and / or the structure property relationship and related advantage of 1D structure on material performance is not clearly transferred into the stated application performance or related to a performance increase. Regarding the first point I suggest to systematically and not only exemplarily combine gas adsorption, TEM and SAXS data for structure analysis and relating observations to the process parameters and solution composition.

Response: We thank the reviewer for these comments. Despite extensive advances in synthetic approaches, the self-assembly of amphiphilic block copolymers into 1D ordered mesoporous nanofibers with tunable mesophases has still not yet been reported. In this case, we have shown the possibility of modulating the monomicelle self-assembly kinetics through a HMT hydrolysis induced process, which provides a new opportunity to realize simultaneous control over both the ordered mesostructure and 1D morphology. This simple but powerful monomicelle-directed self-assembly strategy may bring new inspiration for designing complex structures of functional mesoporous nanomaterials.

For the self-assembly mechanism, we have carried out a series of control experiments to investigate the possible influencing factors on the morphology and porous structure of the prepared materials, including the catalyst, temperature, HMT concentration, and F127-Resol mass ratio. These results clearly demonstrates that this monomicelle system follows a kinetics-mediated 1D oriented self-assembly mechanism. The key feature of our synthesis is the use of HMT molecule as a mediator and curing agent because it is subject to slow hydrolysis into formaldehyde and ammonia at an elevated temperature. Thereby, reasonable control of the reaction temperature with adoption of an appropriate HMT concentration might allow to finely balance the kinetics between monomicelle self-assembly and resol polymerization, so as to simultaneously realize 1D morphological and mesoporous structure controls. SEM and TEM images show that the prepared materials possess 1D fiber-like morphology and mesoporous nanostructure. Meanwhile, the N₂ sorption isotherms reveal the existence of both mesopores and micropores in the formed nanomaterials. Moreover, SAXS patterns further provide evidences that the mesoporous structure is highly ordered and can be adjusted from 3D cubic (*Im-3m*) to 2D hexagonal (*p6mm*) symmetries.

For the application, the advantages of using the 1D ordered mesoporous nanostructure for energy storage and water purification are described as follows: First, the nanoscale diameter and 1D morphology of OMCFs provide much shorter migration distances for mass transportation in comparison with bulk materials, allowing organic molecules and ions to easily penetrate into the whole nanomaterials. Second, the ultra-long OMCFs can form hierarchical webs integrating nanoscale and the microscale morphologies effects, which not only provides a 3D continuous pathway for mass transport but also offers multi-level pore structures and large stacking space to buffer the mechanical stress/strain and volume change during reaction processes. Third, the 3D open ordered mesoporous structure and thin pore walls can offer plentiful confined spaces, surfaces/interfaces and active sites for guest molecule storage and reaction. Fourth, the homogeneous N, O dopants in OMCFs can further alter the electronic conductivity and interfacial wettability, thus additionally contributing to their performance improvement.

In order to further solid the conclusions, we have supplemented a series of control experiments in the revised manuscript. Firstly, we have investigated the effects of pH value and reactant concentration on the morphology and structure of products (Supplementary Fig. 13, 14). Secondly, we have also investigated the mechanical performance between the randomly linked and oriented OMCFs cryogels (Supplementary Fig. 17). Thirdly, we have provided more evidences to demonstrate the advantages of the formed OMCFs in terms of applications such as dye adsorption and water evaporation experiments (Supplementary Fig. 24 and 27). Fourthly, we have summarized some recently reported papers about the carbon-based materials for sodium ion storage and seawater desalination (Supplementary Tab. 2 and 3). Fifthly, we have added more details and statements in the revised the manuscript to describe the corresponding fabrication process and experiment parameters.

Considering the novelties and significances, this unique study has implications for a host of researchers in diverse fields related to material science, electrochemistry, water purification, nanotechnology and self-assembly chemistry. Therefore, we believe that the revised manuscript is of sufficient quality to be published in *Nature Communications*.

Comment 1. *In Figure 1 the authors show nitrogen adsorption data and their pore size distribution analysis. The pore size distribution shows mesopores (IUPAC: 2-50 nm) as well as micropores (IUPAC < 2nm) as far as this can be deduced from the x-axes and adsorption isotherm. This should be included into the discussion as well as the exact analysis of this data (models used etc.). The authors only refer to mesopores within the entire manuscript. The micropore formation should be included into the mechanistic discussion (Figure 4), too. In addition, it has to*

be considered that the presence of micropores would probably significantly increase the specific surface area. Furthermore, the measurement seems not to be finished as the desorption branch is not measured back to low relative pressure. Why is this the case? In this regard as well Figure S11 has to be explained as adsorption and desorption branch are not coming into contact again in the low partial pressure regime. The authors relate the isotherm type to a type IV (I assume a type Iva according to IUPAC) isotherm. This is not fully consistent as the adsorption branch shows a very slow increase and the desorption shows pore blocking and micropores. In addition, another increase at very high relative pressure occurs and should be explained. Thus, the adsorption data interpretation should be reconsidered and gas adsorption results should be added for all relevant porous structures not only for the one example in Figure 1. It should be especially added to Figure 3, to complement TEM and SAXS data. To my opinion, this would improve and solidify the interpretation of porous structure formation together with SAXS data in Figure 3 as well as the mechanistic insights discussed in Figure 4. Furthermore, according to the IUPAC report from 2015 (Pure Appl. Chem. 2015; 87(9-10): 1051–1069), it is known that nitrogen adsorption bears a relatively large error with respect to pore size determination, especially with respect to varying surface chemistry. IUPAC thus recommends argon adsorption. Especially carbon materials are known to adsorb species from the environment and thus change their surface chemistry. Why did the authors use nitrogen and not argon adsorption measurements and why only for two selected examples?

Response: We thank the reviewer for these useful comments. First of all, here we mainly emphasize the creation of mesopores using monomicelles as the building units. Actually, for carbon material, especial those obtained by polymer calcination, micropores are generally present in their skeleton. The micropores are mainly originated from the removal of PEO segments from the pore walls and evolution of gases from the organic polymers during carbonization (*Nat. Commun.* 8, 15020 (2017); *Energy Environ. Sci.* 7, 3574 (2014)). According to the comments of the reviewer, we have added the corresponding statements about the micropores into the revised manuscript. The N₂ sorption isotherms of the OMCs display characteristic type-IV curves with H₄ hysteresis loop, demonstrating the present of micropores and mesopores in OMCs. The specific surface areas were calculated by utilizing the Brunauer-Emmett-Teller (BET) method. The pore size distributions were estimated by using Barrett-Joyner-Halenda (BJH) model. The total pore volumes were calculated at $P/P_0 = 0.995$. It is worth noting that all these data collected are from the adsorption branches of N₂ adsorption isotherms, as they are more likely to reflect the pore structure information than desorption curves. Thus, the range (0.1 - 1.0) of desorption branches measured in the sorption isotherms have no effect on the pore analysis. In original Figure S11, the adsorption and desorption branches are not

coming into contact, probably due to too little sample amount or the presence of some organic matter on the surface of carbon framework. Thus, we have re-measured the N₂ sorption isotherms by extending degassing time (Supplementary Fig. 13d). Moreover, the hysteresis loop at a high pressure ($P/P_0 = 0.89 - 0.98$) reflects the existence of large pores, which may be ascribed to the cross-stacking among the nanofibers. In addition, we have added the N₂ sorption results to solidify the interpretation of porous structure formation (Supplementary Fig. 13). The argon adsorption isotherm can be used to analyze various pores, especially micropores, because of its smaller molecular size, but here we mainly emphasize mesopores, so nitrogen adsorption isotherms is sufficient. As stated in the manuscript, we have used SEM and TEM images to illustrate the 1D mesoporous structure. Also, we have used SAXS data to confirm the mesoporous structure evolution, as shown in Fig 3. In our system, an increase of the F127/Resol mass ratio can expand the monomicelle size and subsequently reduce the micellar curvature to minimize the interfacial energy, which makes the monomicelles fusion easier. As a result, the adjacent micelles would undergo a touching, merging, and fusing process along the long-axis direction, resulting in a mesostructural transfer from 3D cubic (*Im-3m*) to 2D hexagonal (*p6mm*) symmetries. N₂ sorption isotherms were further adopted to indicate the copresence of micropores and mesopores in the OMCs. All these data well demonstrate the formation of 1D ordered mesostructures in this case.

On Page 6, Line 11, in the revised manuscript, we have added the description: “N₂ sorption isotherms of the OMCs is of type-IV curves with H₄ hysteresis loop (Fig. 1g), showing the present of micropores and mesopores in OMCs. The micropores are mainly originated from the removal of PEO segments from the pore walls and evolution of gases from the organic polymers during carbonization.”

On Page 11, Line 14, in the revised manuscript, we have added: “The BET surface areas of the obtained mesoporous nanofibers are calculated to be ~ 452, 421, 347, and 264 m² g⁻¹, respectively (Supplementary Fig. 13). The corresponding pore size distributions are centered at ~ 6.0, 6.8, 7.4, and 8.0 nm, respectively. All data are summarized in Supplementary Tab. 1.”

Accordingly, in the revised Supplementary Information, we have added a new figure and a new table. “**Supplementary Fig. 13.** N₂ sorption isotherms (A) of the mesoporous carbon nanofibers prepared by the kinetically driven monomicelle oriented self-assembly approach using different F127/Resol mass ratios: (a) 0.30, (b) 0.35, (c) 0.40, and (d) 0.45. (B) The corresponding pore size distribution curves.”

“**Supplementary Table 1.** Physicochemical parameters of the resultant mesoporous carbon nanofibers prepared by the kinetically driven monomicelle oriented self-assembly approach using different F127/Resol mass ratios.”

In the revised manuscript, we have added a new reference: “50. Dutta S, Bhaumik A, Wu KC-W. Hierarchically porous carbon derived from polymers and biomass: effect of interconnected pores on energy applications. *Energy Environ. Sci.* 7, 3574-3592 (2014).”

Comment 2. *The authors use the term “F127/HMT/Resol monomicelles”. This term should be defined. Subsequently, the authors state, that these “monomicelles”, which seem to be detrimental for the desired material formation, are formed at a stirring rate of 300 rpm. To make this value reproducible more information about the process such as about the flask / reactor / volumes etc. is needed. A pure stirring rate is not useful. All information should be provided and ideally the process understanding beyond a pure value for a stirring rate under very specific conditions should be provided. The experimental section should be reconsidered with respect to providing all needed information. Figure S1 for example suggests that the hydrothermal reaction was conducted in simple glass vials with blue cap. Is this really the case?*

Response: We appreciate the reviewer for these suggestions. We have accepted them. In our system, the monomicelle is specifically refer to F127/Resol monomicelles. We have modified the term “F127/HMT/Resol monomicelles” to “F127/Resol monomicelles”. Under a stirring rate of 300 rpm, uniform core-shelled spherical F127 aggregations are firstly formed in solution, where PPO domains located at center as the core and surrounded by PEO segment shells. The resol oligomers have plenty of hydroxy groups (-OH) which can interact with the PEO segments of Pluronic F127 through hydrogen bonds, resulting in the well-defined F127/Resol monomicelles. In the Method Section of the Supplementary Information, we have provided more information to describe the synthesis process of the mesoporous carbon nanofibers including the flask / reactor / volumes / temperature, stirring bar *etc.* All of these parameters are important for the formation of stable monomicelles and ultimately self-assembly with resol oligomers into mesoporous nanofibers. The hydrothermal reaction was conducted in a Teflon-lined stainless-steel autoclave. In Supplementary Fig. 1, glass vials are utilized to show the colour evolution of the solution before and after reaction.

On Page 4, Line 5, in the Method Section of the revised Supplementary Information, we have added the description: “Typically, in the first step, 1.0 mL of phenolic resol solution, 0.24 g of Pluronic F127, and 0.08 g of HMT were continuously added into a 200 mL flask with 80 mL of deionized water under stirring rate of 300 rpm. The stirring bar was 2 cm in length. After continuously stirring for 2 h at ambient temperature, the mixture was transferred into a 100 mL Teflon-lined stainless-steel autoclave and heated at 100 °C for other 24 h.”

On Page 7, Line 6, in the revised manuscript, we have added the description: “In the spherical monomicelles, the hydrophobic PPO segments of Pluronic F127 surfactant serves as the core and are surrounded by resol oligomers associated hydrophilic PEO shells to form a core-shell structure as the meso-building block (Fig. 2c).”

Comment 3. *Figure S5 does not clearly show ordered mesoporosity? Could the authors explain this image clearly and relate to the scale bar? In several figure captions in the SI the authors state cubic order of mesopores. Which data supports cubic mesopore arrangement? I do neither see SAXS data showing clear mesopore order in these Figures nor a relation to SAXS data shown elsewhere. In Figure 3 SAXS data is shown but not in all Figures in which a specific order is proposed. On the other hand, gas adsorption data is missing in Figure 3. SAXS and gas sorption data should be added e.g. to Figure S8 for example, too. The statements regarding structural properties should be carefully re-checked and either corrected or supported or linked to the corresponding data. For example, Figure S8 does not even indicate the presence mesopore in the first and last image. In addition, the assignment of figure letters in Figure S8 (a, b, c, d) does not seem to be correct. This makes it in general very difficult to consistently review this manuscript and connect processing parameters and resulting structural properties. Characterization data should be completed, Figures should be correctly assigned and statements should be clearly connected to data. The figure caption in Figure 2 does not fully correspond to the Figure content. It is difficult to understand especially images h and i. For example, what is displayed at the right y-axis in h and how was this value determined? The right y-axis description is not visible in the pdf as it is overlaid by the Figure i. To give one further example: In the discussion referring to Figure 2 the authors state at page 8: “well-defined mesoporous structure (Fig. 2f).” From the depicted SEM images no mesopore structure can be deduced. The entire paragraph should be reconsidered and corrected. Statements should be supported and linked to the corresponding data.*

Response: We thank the reviewer for these helpful comments. Supplementary Fig. 5 is the high-resolution TEM image of the edge of a mesoporous nanofiber, where the tilted sample position and low contrast difference may interfere with the observation of the pores. Thus, some guide lines are introduced to assist in observation, as shown in the revised Supplementary Fig. 5. It can be seen that the mesopores are uniform with a size of ~ 6 nm and the mesopore carbon walls mainly consist of amorphous structure. Meanwhile, the ordered mesoporous structures mentioned in the control experiments are the samples demonstrated in Fig. 1 or Fig. 3, which have been confirmed in detail. As is stated above, we have used SEM and TEM images to illustrate the 1D mesoporous structure. Also, we have used SAXS data to confirm their ordered features. Herein, we have further provided more N₂ sorption isotherms to investigate the mesoporous structures of the carbon nanofibers, as shown in Supplementary Fig. 13. In Supplementary Fig. 8, we have clearly indicated that the products evolved from nonporous nanoparticle, to mesoporous nano-beans, to ordered mesoporous nanofibers, and to irregular solid particle when the HMT

concentration was increased from 0.2 to 1.6 g L⁻¹. The ordered mesoporous structure described in Supplementary Fig. 8c is the same as that of the sample shown in Fig. 1. In order to better observe the morphology and structure of the synthesized materials, we have collected the corresponding magnification images, as shown in the insets of Supplementary Fig. 8. Furthermore, all of the the statements regarding structural properties have been carefully re-checked and either corrected or linked to the corresponding data.

In the revised Supplementary Information, we have modified the caption of Supplementary Fig. 8. “**Supplementary Fig. 8.** The TEM images of the carbon materials prepared by the kinetically driven monomicelle oriented self-assembly approach using different HMT concentrations: (a) 0.2, (b) 0.4, (c) 0.8 and (d) 1.6 g L⁻¹. Insets are the corresponding magnification TEM images. The synthetic processes were carried out similarly to the above Supplementary Method section except that the HMT amount was precisely controlled. The polymeric samples were converted into carbon ones through a carbonization process at 800 °C for 2 h in N₂ atmosphere.”

Accordingly, in the revised Supplementary Information, we have added a new figure and caption. “**Supplementary Fig. 13.** N₂ sorption isotherms (A) of the mesoporous carbon nanofibers prepared by the kinetically driven monomicelle oriented self-assembly approach using different F127/Resol mass ratios: (a) 0.30, (b) 0.35, (c) 0.40, and (d) 0.45. (B) The corresponding pore size distribution curves.”

Comment 4. *Regarding the proposed mechanism (Figure 4): On which data or references is the resol location within the micelles based on? Could the authors make this clear in the manuscript? The observed structural data should be reconsidered with this mechanism e.g. including the role of micropore formation? At page 13 the authors state: “On increasing the temperature (mode II, 100 °C) enables the increase of hydrogen-bonding interaction between the resol oligomers and monomicelles”. This seems counter intuitive to me. Could the authors specify this statement? Following up on the 1D structure formation, the authors report on aerogel formation by an “ice-templating strategy” and specifically describe the resulting structure. Could the authors comment on reproducibility of this aerogel structure and resulting material characteristics such as the capacity or adsorption properties shown in Figure 6? I assume the resulting aerogel structure sensitively depends on the process conditions.*

Response: We thank the reviewer for these useful comments. In our system, the resol oligomers have plenty of hydroxy groups (-OH) which can interact with the PEO segments of Pluronic F127 through hydrogen bonds (*Angew.*

Chem. Int. Ed. 44, 7053 (2005) ; *J. Am. Chem. Soc.* 128, 11652 (2006)). The micropores are mainly originated from the removal of PEO segments from the pore walls and evolution of gases from the organic polymer during carbonization (*Nat. Commun.* 8, 15020 (2017); *Energy Environ. Sci.* 7, 3574 (2014)). Increasing the reaction temperature (mode II, 100 °C) is conducive to the hydrolysis process of HMT molecules, which promotes the release of NH₃ molecules in the solution, and ultimately resulting in faster polymerization rate between resol oligomers. The hierarchical cryogels were prepared by an ice-templating strategy. The ice-templating method has good reproducibility and has been widely used (*Science* 354, 107 (2016) ; *Sci. Adv.* 4, eaat7223 (2018)). When being compared with randomly linked nanofibers, the hierarchical cryogels exhibit greater maximum stress and narrower hysteresis loop, indicating outstanding mechanical performance (Supplementary Fig. 17). As an anode material for sodium ion battery, the hierarchical cryogel delivers a stable capacity of 346 mAh g⁻¹ at the current density of 0.1 A g⁻¹, which is better than that of many carbon-based materials reported previously (Supplementary Tab. 2). Meanwhile, the hierarchical cryogel also shows a high adsorption capacity of ~ 384 mg g⁻¹ for Rhodamine B (Supplementary Fig. 24). In addition, the hierarchical cryogel displays a high evaporation rate of 2.4 kg m⁻² h⁻¹, which is ~ 1.5 times that for randomly linked nanofibers (Supplementary Fig. 27).

On Page 14, Line 4, in the revised manuscript, we have added the description: “Increasing the reaction temperature (mode II, 100 °C) is conducive to the hydrolysis process of HMT molecules, which promotes the release of ammonia molecules in the solution, and ultimately resulting in faster polymerization rate between resol oligomers.”

On Page 17, Line 3, in the revised manuscript, we have added: “When being compared with randomly linked nanofibers cryogel, the oriented OMCFs exhibit greater maximum stress and narrower hysteresis loop (Supplementary Fig. 17), indicating good mechanical robustness.”

On Page 19, Line 19, in the revised manuscript, we have added: “That can be confirmed by time-dependent adsorption tests (Supplementary Fig. 24). The OMCF cryogels possess an impressive adsorption capacity of 384 mg g⁻¹ for Rhodamine B, which is higher than that of the commercial activated carbon.”

Accordingly, in the revised Supplementary Information, we have added three new figures and their captions. “**Supplementary Fig. 17.** Stress-strain curves of (a) randomly linked and (b) oriented OMCFs cryogels under high strain compression, respectively. The inserted SEM images are of the corresponding samples.”

“**Supplementary Fig. 24.** Time-dependent adsorption capacities of Rhodamine B onto the ordered mesoporous carbon nanofibers and commercial activated carbon.”

“**Supplementary Fig. 27.** (a) Mass change curves of the water on the hierarchical OMCFs cryogel and randomly

linked nanofiber cryogel evaporators under one-sun illumination. (b) Schematic illustration showing the water transportation of the above-mentioned two solar-driven evaporators.”

In the revised manuscript, we have added a new reference: “51. Mao L-B, *et al.* Synthetic nacre by predesigned matrix-directed mineralization. *Science* 354, 107-110 (2016). ”

Comment 5. *Furthermore, the described material properties such as dye adsorption are not very surprising but rather reflecting well-known carbon material properties. Could the authors comment on specific advantages over carbon-based materials prepared via alternative and well-known strategies? Is the adsorption capacity significantly increased for example? I cannot deduce such a performance increase which could be related to the 1D structure from the given data. The authors indicate such improved performance e.g. by the statement: “...exhibits excellent cyclability with a stable capacity of 346 mAh g⁻¹ after 100 cycles (Supplementary Fig. S16), which is much better than that of the nonporous carbon particle electrode (Supplementary Fig. S17), indicating the morphological and structural advantages of the OMCFs.”. But no reference to the state of the art and most recent literature is given and a clear structure property relationship is not deduced. Consequently, this statement remains vague and imprecise. In a second example on water filtration the authors state:” peaks of organic dyes at 552 and 623 nm almost completely disappeared (Fig. 6i), indicating superior filtering performance of the nanofilter device.”. Without giving adsorption capacity and comparison to benchmark materials this statement does not allow to deduce and superior performance. Furthermore, which functional groups are responsible for high adsorption capacity in case the material exists of almost pure carbon? In general, none of the applications, which are all known for classic carbon materials are clearly related to the 1D material design. In general this application-related section remains weak with respect to innovative performance or structure driven insights but rather shows expected behavior of carbon-based materials.*

Response: We appreciate the reviewer for these useful comments. The OMCF cryogel possesses an impressive adsorption capacity of 384 mg g⁻¹ for Rhodamine B, which is higher than that of commercial activated carbon (Supplementary Fig. 24). The superior adsorption performance of the nanofilter devices is attributed to their high surface area, open mesostructure, π - π structure and N, O-doped properties. Furthermore, it is clear that the vapor device with hierarchical cryogel presents a high evaporation rate of 2.4 kg m⁻² h⁻¹, which is ~ 6 and 1.5 times that for pure water and randomly linked nanofibers, respectively (Supplementary Fig. 27). In addition, compared with results reported in recent literature (Supplementary Tab. 2, 3), both sodium ion storage and water evaporation performances

of the OMCFs are better than that of many other carbon-based materials reported previously. The 1D morphology and pore structure stability of OMCFs can be confirmed by TEM analysis after a long-term cycle (Supplementary Fig. 21).

In summary, the exceptional performance of OMCFs can be explained with their unique 1D morphology, ordered mesostructure and rich N-doped chemical property. First, the nanoscale diameter and 1D morphology of OMCFs provide much shorter migration distances for mass transportation in comparison with bulk materials, allowing organic molecules and ions to easily penetrate into the whole nanomaterials. Second, the ultra-long OMCFs can form hierarchical webs integrating nanoscale and the microscale morphologies effects, which not only provides a 3D continuous pathway for mass transport but also offers multi-level pore structure and large stacking space to buffer the mechanical stress/strain and volume change during reaction processes. Third, the 3D open ordered mesoporous structure and thin pore walls can offer plentiful confined spaces, surfaces/interfaces and active sites for guest molecule storage and reaction. Fourth, the homogeneous N, O dopants in OMCFs can further alter the electronic conductivity and interfacial wettability, thus additionally contributing to their performance improvement.

On Page 17, Line 18, in the revised manuscript, we have added: “Such cycling performance is better than that of many other carbon-based electrodes reported previously (Supplementary Tab. 2). The pore structure stability of OMCFs can be confirmed by TEM analysis after a long-term cycle (Supplementary Fig. 22).”

On Page 19, Line 19, in the revised manuscript, we have added: “That can be confirmed by time-dependent adsorption tests (Supplementary Fig. 24). The OMCF cryogels possess an impressive adsorption capacity of 384 mg g⁻¹ for Rhodamine B, which is higher than that of commercial activated carbon.”

On Page 20, Line 5, in the revised manuscript, we have added: “It is clear that the vapor device with hierarchical OMCFs cryogels presents a high evaporation rate of 2.4 kg m⁻² h⁻¹, which is ~ 6 and 1.5 times that for pure water and randomly linked nanofibers, respectively (Supplementary Fig. 27).”

On Page 20, Line 12, in the revised manuscript, we have added: “The exceptional performance of OMCFs for practical application can be explained with their unique 1D morphology, ordered mesostructure and rich N-doped chemical property. First, the nanoscale diameter and 1D morphology of OMCFs provide much shorter migration distances for mass transportation in comparison with bulk materials, allowing organic molecules and ions to easily penetrate into the whole nanomaterials. Second, the ultra-long OMCFs can form hierarchical webs integrating nanoscale and the microscale morphologies effects, which not only provides a 3D continuous pathway for mass transport but also offers multi-level pore structures and large stacking space to buffer the mechanical stress/strain and

volume change during reaction processes. Third, the 3D open ordered mesoporous structure and thin pore walls can offer plentiful confined spaces, surfaces/interfaces and active sites for guest molecule storage and reaction. Fourth, the homogeneous N, O dopants in OMCFs can further alter the electronic conductivity and interfacial wettability, thus additionally contributing to their performance improvement.”

Accordingly, in the revised Supplementary Information, we have added three new figures and their captions. “**Supplementary Fig. 17.** Stress-strain curves of (a) randomly linked and (b) oriented OMCFs cryogels under high strain compression, respectively. The inserted SEM images are of the corresponding samples.”

“**Supplementary Fig. 24.** Time-dependent adsorption capacities of Rhodamine B onto the ordered mesoporous carbon nanofibers and commercial activated carbon.”

“**Supplementary Fig. 27.** (a) Mass change curves of the water on hierarchical OMCFs cryogel and randomly linked nanofibers cryogel evaporators under one-sun illumination. (b) Schematic illustration showing the water transportation of the above-mentioned two solar-driven evaporators.”

Furthermore, in the revised Supplementary Information, we have added two new tables. “**Supplementary Table 2.** Comparison of the electrochemical performances of the ordered mesoporous carbon nanofiber electrode in this work with some carbon-based electrodes reported in the literature.”

“**Supplementary Table 3.** Comparison of the water evaporation rate of the ordered mesoporous carbon nanofiber cryogels in this work with some carbon-based evaporators reported in the literature.”

REVIEWER COMMENTS

Reviewer #1 (Remarks to the Author):

After carefully reading the revised manuscript and the authors' rebuttal letter, I arrived to the following conclusions. First, the general message and all the specific comments of my first review report are very well aligned with Reviewer#3's report. Both of us asked for additional experiments or at least the systematic re-evaluation of the data for supporting the scientific claims of the manuscript. While additional experimental data has been provided, mainly by following my advice, I agree with Reviewer#3's original opinion, that the interpretation of the compiled sets of data is generally superficial. Unfortunately, I do not see a major improvement in this point in the new version of the manuscript. After reading the answers and the modifications in response to Reviewer#3's specific points, I think these still do not satisfy the need to systematically explain the designable tuning of the materials properties and the underlying mechanism. According to these observations, I still uphold the opinion that the scientific claims and conclusions are still not fully supported by the presented data and their evaluation. The final structures of the materials cannot be systematically linked to the fabrication conditions. However, this could be repaired by the reorganization of the presented results and their discussion.

Reviewer #2 (Remarks to the Author):

The authors have clarified the points raised by the referees. The quality of the manuscript has been improved. Therefore, I would recommend its publication on Nature Communications.

Reviewer #3 (Remarks to the Author):

To my opinion, the authors addressed all comments satisfactorily. The manuscript improved significantly. The innovation was made clear. I recommend to publish the revised manuscript.

Title: “One-dimensionally oriented self-assembly of ordered mesoporous nanofibers featuring tailorable mesophases via kinetic control”

Point-to-Point Response to Reviewers

Reviewer #1:

After carefully reading the revised manuscript and the authors' rebuttal letter, I arrived to the following conclusions. First, the general message and all the specific comments of my first review report are very well aligned with Reviewer#3's report. Both of us asked for additional experiments or at least the systematic re-evaluation of the data for supporting the scientific claims of the manuscript. While additional experimental data has been provided, mainly by following my advice, I agree with Reviewer#3's original opinion, that the interpretation of the compiled sets of data is generally superficial. Unfortunately, I do not see a major improvement in this point in the new version of the manuscript. After reading the answers and the modifications in response to Reviewer#3's specific points, I think these still do not satisfy the need to systematically explain the designable tuning of the materials properties and the underlying mechanism. According to these observations, I still uphold the opinion that the scientific claims and conclusions are still not fully supported by the presented data and their evaluation. The final structures of the materials cannot be systematically linked to the fabrication conditions. However, this could be repaired by the reorganization of the presented results and their discussion.

Response: We thank the reviewer for these useful comments. Actually, the Reviewer#3 have fully accepted our revision and recommended our manuscript to publish in *Nature Communications*. Even so, we accepted this Reviewer's suggestions and further systematically explained the presented results and offered more detail discussion. Firstly, we have interpreted the underlying reasons of the presented experimental results point-by-point in terms of catalyst, HMT concentration, F127/Resol mass ratio, temperature, and pH value etc. Secondly, we have further expounded the 1D oriented self-assembly mechanism, highlighted the key points, and refined the details. Thirdly, we have systematically summarized the structure and chemical advantages of the resultant produce for practical applications. Fourthly, we also have cited some important references into the revised manuscript to solid the proposed mechanism and conclusion.

According to the comments of the reviewer, we have carefully revised our manuscript again.

In the new revised manuscript, on Page 7, Line 6, we have modified the descriptions as follows: “In the spherical monomicelles, the hydrophobic PPO segments of Pluronic F127 surfactants spontaneously aggregate together due to

the van der Waals interaction and are surrounded by hydrophilic PEO parts to form the core-shell structure (Fig. 2c). The resol oligomers can interact with the hydrophilic PEO segments of Pluronic F127 surfactants through hydrogen bonding.”

On Page 7, Line 19, we have modified the description as follow: “Meanwhile, the pH value of the solution gradually increased to 9.5 in the initial 3h due to the thermally induced decomposition of HMT molecules, and then remained unchanged (Fig. 2i), demonstrating the buffering role of HMT molecules.”

On Page 9, Line 1, we have added the descriptions as follows: “The variability in the products confirms that the used catalyst has a strong influence of the monomicelle self-assembly kinetics. Compared with ammonia and NaOH, HMT molecules can *in-situ* release of ammonia into the solution and thus dynamically mediate the monomicelle self-assembly kinetics to achieve 1D oriented self-assembly.”

On Page 9, Line 9, we have added the description as follows: “The structural evolution of products is probably because the HMT concentration can remarkably influence the self-assembly behavior between the resol oligomers and F127/Resol monomicelles.”

On Page 9, Line 19, we have added the description as follows: “This is because the reactant concentration can significantly affect the nucleation rate of the resol oligomers and monomicelle self-assembly kinetics.”

On Page 10, Line 9, we have modified the description as follows: “These results indicate that the weakly alkaline environment is favorable for achieving kinetic balance between the resol oligomers polymerization and F127/Resol monomicelles self-assembly to generate 1D morphology and mesoporous structure.”

On Page 11, Line 6, we have modified the description as follows: “It is indicated that the reaction temperature applied can significantly influence the decomposition rate of HMT molecules, thus changing the self-assembly behavior of F127/Resol monomicelle. Meanwhile, too high reaction temperature can lead the monomicelle unstable and even broken.”

On Page 13, Line 22, we have modified the descriptions as follows: “The key feature of our synthesis is the use of HMT molecules as a mediator and curing agent because these can *in-situ* decompose to formaldehyde and ammonia molecules at an elevated temperature. On the one hand, the as-derived ammonia molecules can serve as a pH buffer to intelligently control the self-assembly kinetics of monomicelles on demand. On the other hand, the formaldehyde molecules can act as a mediator to manipulate the molecular interaction between the resol oligomers through hydrogen bonding, that is, to regulate their polymerization kinetics.”

On Page 15, Line 6, we have added the description as follows: “These results indicate that only the *in-situ* release

of ammonia and formaldehyde molecules decomposed from HMT can form ordered 1D mesoporous structure.”

On Page 15, Line 8, we have modified the description as follows: “This is because increasing the amphiphilic Pluronic F127 mass ratio would expand the monomicelle size and subsequently decrease the curvature of the micellar surface to minimize the interfacial energy, which allows the monomicelles to form cylindrical nanorods rather than nanospheres.”

On Page 15, Line 15, we have modified the descriptions as follows: “However, a higher F127/Resol mass ratio could cause a change in micellar morphology and consequently induce mesoporous nanofibers to be bend and increase in diameter (Supplementary Fig. 14 d). In addition, further increasing the reaction temperature (mode III, 140 °C) could break down the overall kinetic equilibrium between the monomicelles self-assembly and resol oligomers polymerization (Supplementary Fig. 15c), resulting in nonporous solid particles. Therefore, well-defined 1D mesostructures with precisely tunable mesophases can be facilely synthesized through a self-assembly kinetic control pathway, which could be promising for diverse applications.”

On Page 21, Line 3, we have modified the descriptions as follows: “The exceptional performance of OMCFs for practical application can be attributed to their unique nanostructure and chemical properties, including nanoscale dimension, 1D morphology, high aspect ratio, 3D ordered open mesostructure, amorphous framework, and rich O, N-dopants. Firstly, the nanoscale dimension and 3D ordered open mesostructure of OMCFs offer much shorter migration distances for mass transfer in comparison with bulk materials, allowing organic molecules and metal ions to easily penetrate into the whole nanomaterials. Secondly, the 1D morphology and high aspect ratio of OMCFs can construct hierarchical porous networks integrating nanoscale and the microscale morphologies effects, which not only provides a 3D continuous pathway for mass transfer and electron transport but also offers multi-level pore structures and large stacking space to buffer the mechanical stress/strain and volume change during reaction processes. Thirdly, the big pore size, thin pore walls, and amorphous frameworks can offer large confined spaces, accessible surfaces/interfaces and abundant active sites for guest molecule storage and reaction. Fourthly, the homogeneous rich N, O-dopants in OMCFs can further enhance the electronic conductivity and tailor interfacial wettability, thus additionally advance their performance.”

Accordingly, in the new revised Manuscript, we have added some references: “51. Monteiro MJ, de Barbeyrac J. Free-Radical Polymerization of Styrene in Emulsion Using a Reversible Addition-Fragmentation Chain Transfer Agent with a Low Transfer Constant: Effect on Rate, Particle Size, and Molecular Weight. *Macromolecules* 34, 4416-4423 (2001).

52. Zhang F, et al. A Facile Aqueous Route to Synthesize Highly Ordered Mesoporous Polymers and Carbon Frameworks with Ia $\bar{3}d$ Bicontinuous Cubic Structure. *J. Am. Chem. Soc.* 127, 13508-13509 (2005).

53. Owen SC, Chan DPY, Shoichet MS. Polymeric micelle stability. *Nano Today* 7, 53-65 (2012).

54. Peng L, et al. Versatile nanoemulsion assembly approach to synthesize functional mesoporous carbon nanospheres with tunable pore sizes and architectures. *J. Am. Chem. Soc.* 141, 7073-7080 (2019).

55. Ringsdorf H, Schlarb B, Venzmer J. Molecular architecture and function of polymeric oriented systems: models for the study of organization, surface recognition, and dynamics of biomembranes. *Angew. Chem. Int. Ed.* 27, 113-158 (1988).”

Reviewer #2:

Recommendation: Publish in Nature Communications.

The authors have clarified the points raised by the referees. The quality of the manuscript has been improved.

Therefore, I would recommend its publication on Nature Communications.

Response: We thank the reviewer very much for the approval of this work.

Reviewer #3:

Recommendation: Publish in Nature Communications.

To my opinion, the authors addressed all comments satisfactorily. The manuscript improved significantly. The innovation was made clear. I recommend to publish the revised manuscript.

Response: We thank the reviewer very much for the approval of this work.

REVIEWERS' COMMENTS

Reviewer #1 (Remarks to the Author):

The article has significantly been improved. (Some typos were introduced that can be corrected in the proof.)